# A roadmap for ribosome assembly in human mitochondria

Elena Lavdovskaia [1,2,3,10], Elisa Hanitsch[2,10], Andreas Linden[4,10], Martin Pašen[5], Venkatapathi Challa[1,2], Yehor Horokhovskyi [5], Hanna P. Roetschke[5,6,7], Franziska Nadler[2], Luisa Welp[4,8], Emely Steube[2], Marleen Heinrichs[1,2], Mandy Mong-Quyen Mai[2], Henning Urlaub [3,4,8,9] ✉, Juliane Liepe [5] ✉ & Ricarda Richter-Dennerlein [1,2,3,9] ✉

Mitochondria contain dedicated ribosomes (mitoribosomes), which synthesize the mitochondrial-encoded core components of the oxidative phosphorylation complexes. The RNA and protein components of mitoribosomes are encoded on two different genomes (mitochondrial and nuclear) and are assembled into functional complexes with the help of dedicated factors inside the organelle. Defects in mitoribosome biogenesis are associated with severe human diseases, yet the molecular pathway of mitoribosome assembly remains poorly understood. Here, we applied a multidisciplinary approach combining biochemical isolation and analysis of native mitoribosomal assembly complexes with quantitative mass spectrometry and mathematical modeling to reconstitute the entire assembly pathway of the human mitoribosome. We show that, in contrast to its bacterial and cytosolic counterparts, human mitoribosome biogenesis involves the formation of ribosomal protein-only modules, which then assemble on the appropriate ribosomal RNA moiety in a coordinated fashion. The presence of excess protein-only modules primed for assembly rationalizes how mitochondria cope with the challenge of forming a protein-rich ribonucleoprotein complex of dual genetic origin. This study provides a comprehensive roadmap of mitoribosome biogenesis, from very early to late maturation steps, and highlights the evolutionary divergence from its bacterial ancestor.

Mitochondria provide the majority of cellular energy by oxidative phosphorylation (OXPHOS). The core subunits of the OXPHOS complexes are encoded by the mitochondrial genome (mtDNA), which requires a dedicated expression apparatus including mitochondrial ribosomes (mitoribosomes). Like the OXPHOS complexes, the mitoribosome represents a multimeric machinery of dual genetic origin. While the ribosomal RNA in the mitochondria (mt-rRNA) is encoded by the mtDNA, all 82 mitoribosomal proteins (MRPs) are encoded in the nucleus, translated in the cytosol and imported into mitochondria. The 55S human mitoribosome is formed by a 39S large mitoribosomal subunit (mtLSU), comprising 52 MRPs, the 16S mt-rRNA and the transfer RNA binding Val (tRNA$^{Val}$), and a 28S small mitoribosomal subunit (mtSSU), containing 30 MRPs and the 12S mt-rRNA[1,2]. How these macromolecular complexes assemble is poorly understood. The evolution of the human mitoribosome was accompanied by a notable increase in its protein mass and a decrease in rRNA content, leading to a different composition and remodeled structure compared to its bacterial counterpart. Knowledge of the well-studied bacterial ribosome assembly pathway cannot,

A full list of affiliations appears at the end of the paper. ✉e-mail: henning.urlaub@mpinat.mpg.de; juliane.liepe@mpinat.mpg.de; ricarda.richter@med.uni-goettingen.de

therefore, simply be extrapolated to mitoribosomes, although they derived from a common ancestor. The essentiality of correct mitoribosome production is highlighted by numerous persons suffering from mitochondrial diseases associated with mutations in genes encoding for mt-rRNA, MRPs or assembly factors, which facilitate the correct maturation and folding of RNAs and positioning of MRPs[3–5]. Recent high-resolution structural snapshots of late mitoribosome assembly intermediates have provided important initial insights into the complex process of mitoribosome maturation and the molecular functions of associated biogenesis factors[6–14]. However, the compositions of earlier biogenesis modules are not known because of their small size, dynamic nature and the challenges with their isolation. First attempts to explore the biogenesis of mitoribosomes biochemically were limited to a pulse SILAC (stable isotope labeling by amino acids in cell culture) approach in which the appearance of newly synthesized MRPs in fully assembled mitoribosomes, which were isolated under harsh conditions, was characterized[15]. However, the incorporation of MRPs into smaller complexes or into the mtLSU or mtSSU could not be determined in that study, in part because of inattention to mass spectrometry (MS) data normalization, restricted quantitative comparison of abundances of all MRPs and a lack of consideration of association and turnover rates. The limited ability to assign MRPs to assembly intermediates using this approach has led to contradictory models that have been controversially discussed in the field[16,17].

Here, we applied an integrated triple-SILAC MS approach with biochemical experiments and mathematical modeling to monitor the sequential incorporation of individual MRPs into biogenesis modules in vivo and created a comprehensive map for human mitoribosome assembly. Results were validated by (1) immunoisolation of the distinct submodules followed by density gradient centrifugation to separate the native complexes and to confirm the composition of the individual modules and (2) assembly perturbation by mt-rRNA or MRP ablation to follow the consequences of loss of function. Our analyses reveal the formation of preassembled protein-only modules, which are available in excess and serve as primed building blocks for ribosome biogenesis. Mathematical modeling allowed the creation of a kinetic model for the mtSSU assembly, providing a framework for determining the impact of changes in kinetic assembly rates of a single MRP or an MRP cluster on the overall mtSSU abundance.

## Stabilities of MRPs versus assembly factors

Before investigating the turnover of mitoribosome assembly intermediates, we first assessed the global intracellular turnover of MRPs and assembly factors in whole-cell lysate and in isolated mitochondria (Extended Data Fig. 1a). HEK293 cells were pulse-labeled with 'heavy' (H) amino acids ([$^{13}C$]$_6$[$^{15}N$]$_4$Arg, +10 Da (Arg-10); [$^{13}C$]$_6$[$^{15}N$]$_2$Lys, +8 Da (Lys-8)) and then chased with 'medium' (M) amino acids ([$^{13}C$]$_6$Arg, +6 Da (Arg-6); [$^2H$]$_4$Lys, +4 Da (Lys-4)) for 12 h, followed by a second chase with unlabeled, 'light' (L) amino acids (Arg-0; Lys-0) for 24 h. The chase time points were chosen on the basis of the reported average protein half-life in human cell lines, which is approximately 46 h (ref. 18). This triple-SILAC MS approach allowed tracking of the exchange of H proteins by M and L proteins. While many proteins follow an exponential decay in their turnover[19,20], the turnover of proteins that undergo stable complex formation is more likely represented by a nonexponential decay[21]. Therefore, we implemented and compared a one-state model and a two-state model[21] to dissect the turnover of single MRPs and MRPs incorporated into mitoribosomes, as well as the turnover of assembly factors (Extended Data Fig. 1b–d, Supplementary Data 1, Supplementary Table 1 and Supplementary Fig. 1). The observed turnover of the total cellular fraction of MRPs and assembly factors was similar to that of the mitochondrial fraction; this is likely because the mitochondrion-resident proteome represents the vast majority of the total cellular fraction of these proteins, suggesting their immediate import as single entities after or during translation. On the other

hand, 37% of all MRPs were better described by the two-state model and showed an initial fast turnover rate, presumably reflecting unbound proteins, and a later slower turnover phase, likely indicating higher stability of the proteins because of complex formation. Furthermore, the turnover rates of the remaining 63% of MRPs better depicted by the one-state-model were comparable to the slower turnover rates of MRPs better explained by the two-state-model, indicating that MRPs incorporated into mitoribosomes have the same turnover. In contrast, assembly factors were largely explained by the one-state-model (73%) and had higher variation in turnover rates than MRPs, indicating a transient interaction rather than stable incorporation into mitoribosome complexes (Extended Data Fig. 1e,f and Supplementary Data 1).

## Reconstructing in vivo mitoribosome assembly

To quantitatively monitor the formation of mitoribosome assembly complexes in vivo, we applied a pulse–chase triple-SILAC approach (pulse: Arg-10 and Lys-8, H; chase: Arg-6 and Lys-4, M), separated mitochondrial lysates including mitoribosome complexes by sucrose density gradient ultracentrifugation and monitored the differentially labeled proteins in isolated fractions by MS over time (Fig. 1a and Methods). Purified 55S mitoribosomes from cells grown in standard L medium were used as an internal standard and spiked into each fraction before MS analysis; this allowed for normalization of H and M signals to the L standard and, therefore, quantitative comparison of abundances of all MRPs in each fraction (Fig. 1b,c, Supplementary Data 2–5 and Supplementary Table 2). The labeling kinetics of MRPs were characterized by a decrease in the H signal and an increase in the M signal intensities over time, demonstrating the substitution of pre-existing MRPs (H labeled) by newly synthesized MRPs (M labeled) (Supplementary Fig. 2 and Supplementary Table 2). The exchange rate did not follow the same magnitude over the gradient fractions. In low-density fractions, the equilibrium between H labeled and M labeled MRPs was reached generally in less than 3 h of chase. By contrast, the H pool predominated in fractions corresponding to the mature subunits even after 12-h chase, indicating the presence of stable complexes with low turnover rates (Supplementary Fig. 2, Supplementary Data 4 and 5 and Supplementary Table 2).

The group of mtSSU MRPs had its maximal abundance in gradient fractions 6–7 where mature 28S mtSSUs sedimented, whereas mtLSU MRPs peaked in fractions 8–9, reflecting the presence of the mature 39S mtLSUs. Both MRP groups were detected in fractions 11–12, where the complete 55S mitoribosome migrated (Fig. 1c). We observed an accumulation of most MRPs in low-density gradient fractions with a tendency to peak in fraction 2 or 3, likely representing subassemblies as observed previously[22–24]. To derive details of the mtSSU and mtLSU assembly pathways, we aimed to cluster the MRPs by their turnover kinetics across sucrose gradient fractions taking into account the assembled structure of the mtSSU and mtLSU (Fig. 1b). Therefore, we first aimed to simplify the complex dataset, that is, reduce the data dimensionality of the normalized abundances of H labeled and M labeled MRPs across fractions and over time (Fig. 1b, step 1 and Supplementary Fig. 2) by computing steady-state abundances and fluxes of MRPs across sucrose gradient fractions (Fig. 1b, step 2, Supplementary Table 3, Supplementary Data 6 and 7, Supplementary Fig. 3 and Methods, 'Flux estimation of MRPs through sucrose gradient fractions'). MRP fluxes across fractions reflect the rates of MRP turnover and transfer across fractions. Estimated fluxes and steady-state abundances across fractions of a given target MRP were then compared to fluxes and abundances of all MRPs that could interact with the target MRP (Fig. 1b, steps 3–4, Supplementary Fig. 4 and Supplementary Data 8 and 9), thus allowing us to derive MRP clusters and generate first mtSSU and mtLSU assembly maps (Fig. 1b). We evaluated each MRP cluster by computing cluster heterogeneity and comparing all alternative clusters that could be derived solely using structural constraints (Fig. 1b, step 5 and Fig. 2). If the heterogeneity

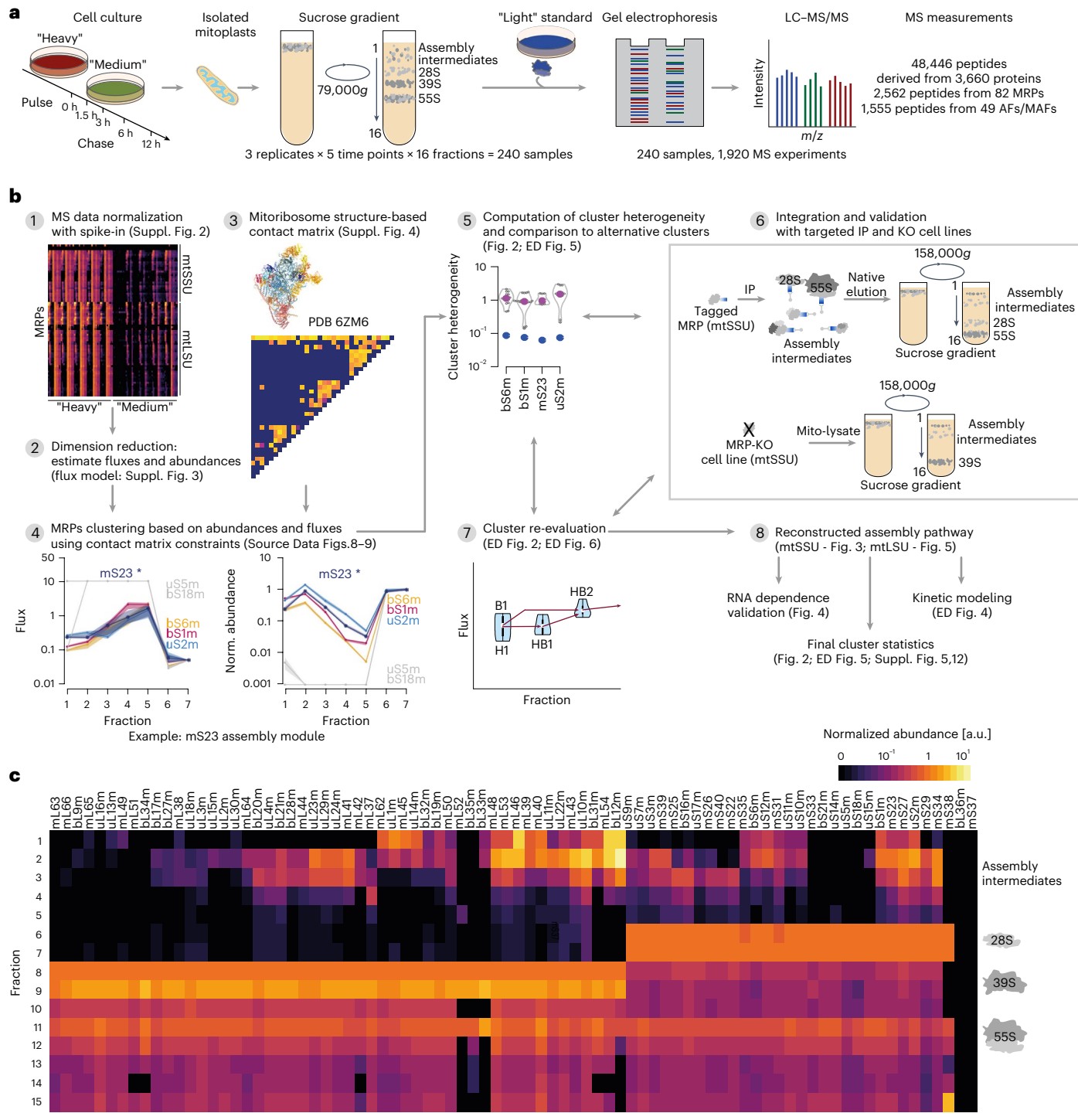

**Fig. 1 | Triple-SILAC experimental design and data analysis summary.**
**a**, Overview of the experimental workflow. HEK293 cells were pulse-labeled with H amino acids (Arg-10 and Lys-8; red) and then chased with M amino acids (Arg-6 and Lys-4; green) for indicated time intervals. Mitoribosomal complexes were separated by sucrose gradient ultracentrifugation (low-resolution' gradient, 79,000g for 15 h). Isolated fractions were spiked with an L standard (isolated 55S mitoribosomes, blue; Arg-0 and Lys-0) and analyzed by LC–MS/MS (*n* = 3). **b**, Schematic of the data analysis workflow to reconstruct mtSSU and mtLSU assembly pathways. Illustrated are all essential steps with references to more detailed figures. IP, immunoprecipitation; KO, knockout. **c**, Normalized MRP steady-state abundance across sucrose gradient fractions. Normalized protein abundance is indicated as a range from black (zero) to light yellow (maximal value). MRPs are arranged on the basis of a hierarchical clustering of abundances across all sucrose gradient fractions. AF, mitoribosome assembly factor; MAF, mitoribosome-associated factor.

of a selected cluster was not the smallest compared to alternative clusters, we biochemically challenged the composition of these clusters and the interdependency between individual MRPs of the module by (1) targeted immunoprecipitation using selected MRPs followed by

'high-resolution' gradient centrifugation with better complex separation in less dense fractions (Methods) and (2) exploring the effects of MRP loss or mt-rRNA depletion on complex formation (Fig. 1b, steps 6–7, Fig. 2 and Supplementary Tables 4 and 5). For example, mS23

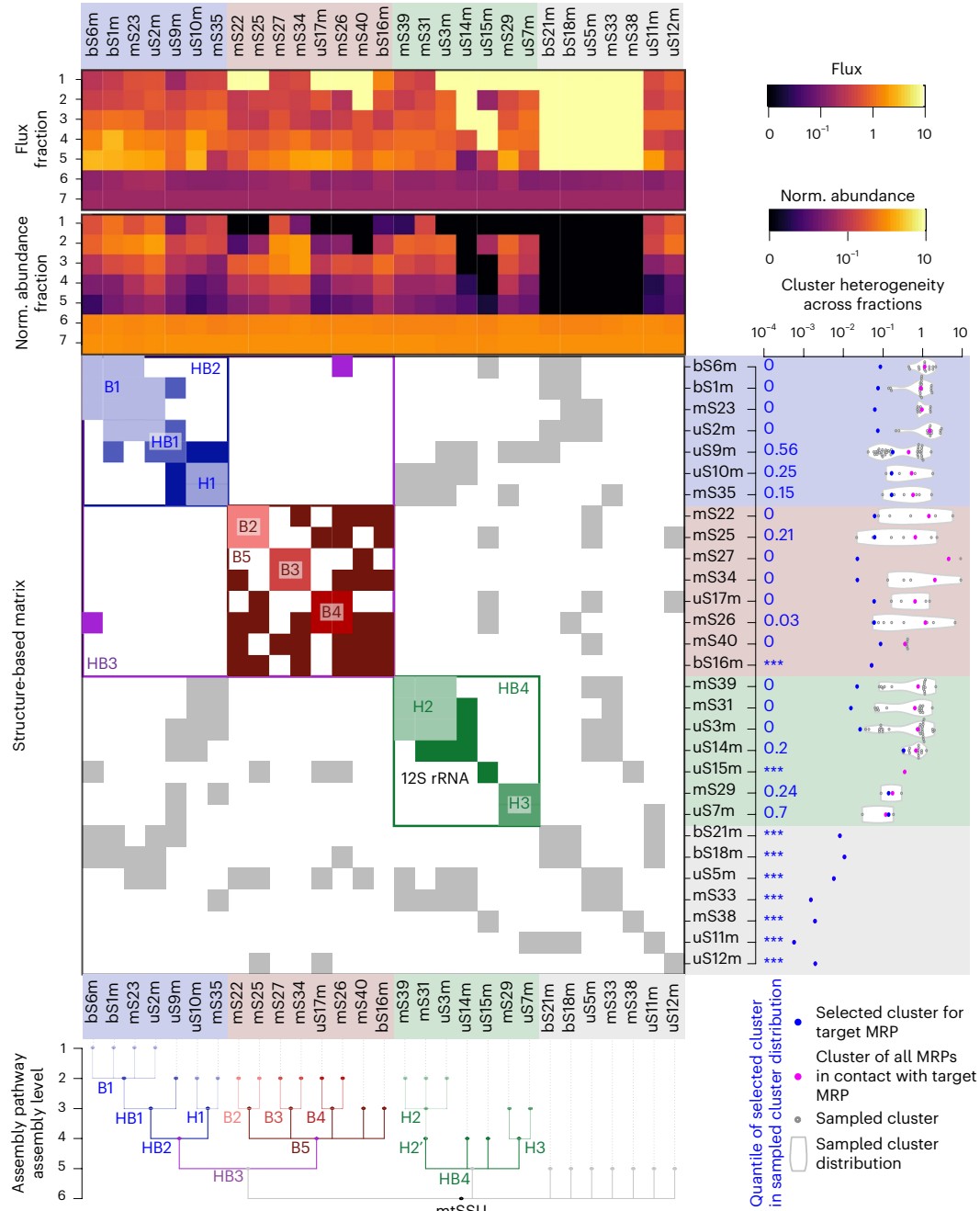

**Fig. 2 | Cluster analysis of mtSSU assembly pathway.** Top, the fluxes and normalized steady-state abundances of all MRPs are shown as a heat map. The mtSSU MRPs are aligned according to their clusters. Center, contact matrix showing all pairwise MRP contacts derived from the known structure of the assembled mtSSU, colored according to their assigned assembly module. Right, cluster heterogeneity for each target MRP indicated as blue dots and compared to the heterogeneity of all possible clusters based only on contact matrix constraints (gray dots and violin plots). Blue numbers define the quantile of the selected target cluster (blue dot) within the alternative cluster distribution, where 0 indicates that the selected target cluster has the lowest heterogeneity and *** indicates the absence of alternative clusters. The resulting assembly pathway is shown as a dendrogram with indicated assembly levels for all MRPs and mtSSU modules.

interacted with five MRPs (bS6m, bS1m, uS2m, uS5m and bS18m) in the assembled mtSSU structure. The MRPs bS6m, bS1m and uS2m showed similar fluxes and abundances across sucrose fractions 1–7 to mS23, suggesting the formation of an early assembly module (Fig. 1b, step 4 and Supplementary Data 8). However, uS5m and bS18m do not belong to the same module because they were only detected in fractions 6 and 7. The selected cluster for mS23 showed the smallest heterogeneity compared to alternative clusters (Fig. 1b, step 5, Fig. 2, and Supplementary Table 4) and was confirmed with targeted

immunoprecipitation (Fig. 1b, step 6 and Extended Data Fig. 3e,f). As a counter example, uS10m was clustered with mS35 in fraction 1, despite a cluster with mS31 and uS9m resulting in lower cluster heterogeneity on the basis of MS data alone (Fig. 2 and Supplementary Table 4). However, targeted immunoprecipitation confirmed that uS10m interacts with mS35 during early assembly steps but not with the remaining MRPs (Extended Data Fig. 3f).

These iterative and complementary experiments allowed us to derive early and late assembly steps of the mtSSU and mtLSU.

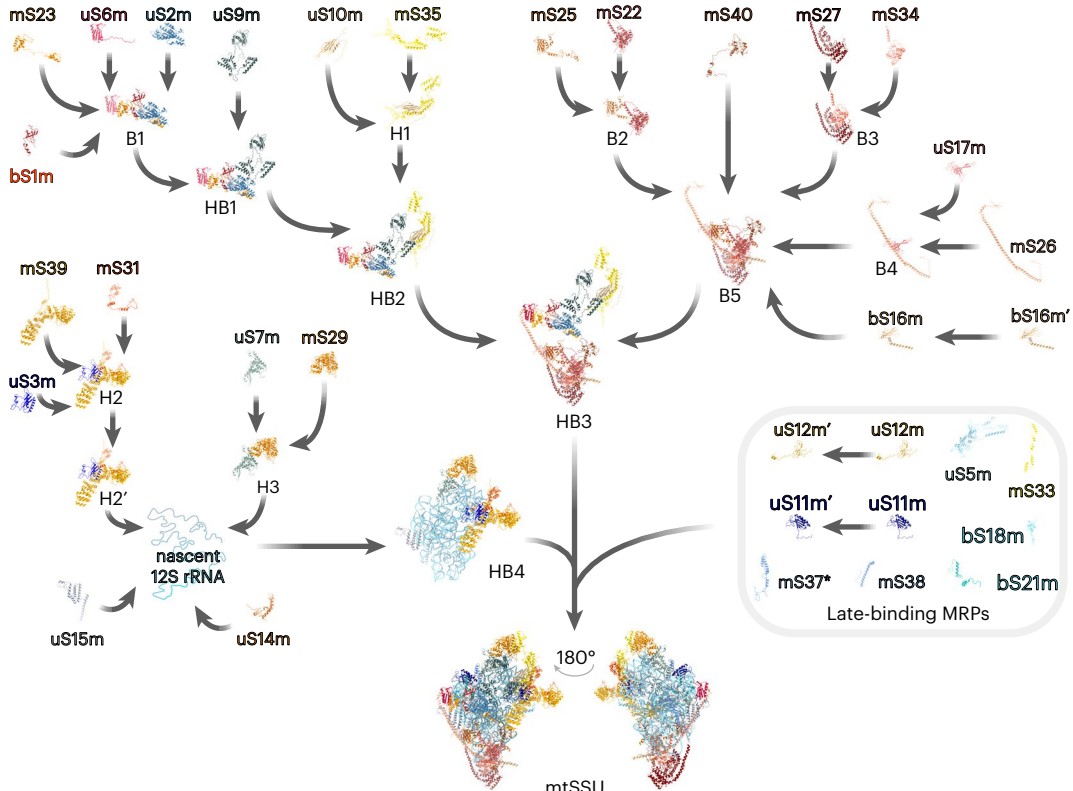

**Fig. 3 | Reconstructed in vivo pathway of the mtSSU assembly.** The biogenesis of the 28S mtSSU involves the sequential incorporation of preassembled MRP clusters to the 12S rRNA moieties. Eight MRPs bind to the maturing mtSSU as individual proteins in the final stages of assembly to fine-tune established biogenesis intermediates. H, mtSSU head; B, mtSSU body; HB, mtSSU head–body assembly module. The prime symbol indicates differences in the kinetic properties of the MRPs and assembly module with the identical MRP composition. *Although not continuously detected in our data set, mS37 presents the last assembling MRP according to structural analysis of late mtSSU assembly intermediates[6].

## The assembly pathway of the mtSSU

The mtSSU contains the conserved decoding center. The folding of the decoding center is one of the final maturation steps during mtSSU assembly; it is facilitated by the concerted action of assembly factors such as ERAL1, MTG3, TFB1M, METTL15 and mtRBFA and further compacted by the association of late-binding MRPs[6,7]. Our work focuses more on the upstream events and sheds light on the so far undefined steps during early assembly. The clustering analysis, considering the mtSSU structural constraints, revealed four major assembly clusters involved in the formation of the mtSSU: B5 (MRPs from mtSSU body), HB2 (MRPs spanning the mtSSU head and body), HB3 (a large protein-only subassembly formed by B5 and HB2) and a 12S mt-rRNA-containing cluster HB4 (Figs. 2 and 3, Extended Data Fig. 2 and Supplementary Fig. 5).

Assembly of the B5 cluster starts with the formation of three modules B2–B4; association of the foot MRPs mS27 and mS34 (B3) initiates cluster biogenesis, which is followed by recruitment of mS22–mS25 (B2) and uS17m–mS26 (B4), with mS40 and bS16m then finally joining. The hierarchical formation of these subassemblies was confirmed by coimmunoprecipitation using selected B5 constituents as baits followed by complex separation using high-resolution sucrose gradient centrifugation (Extended Data Fig. 3a–c). Coisolated components of the B5 complex comigrated in less dense fractions, while MRPs, not belonging to the B5 module, were only observed in fractions 10–11 corresponding to the 28S mtSSU. Conversely, B5 constituents were not detectable in less dense fractions but only in fractions 10–16 when copurified using components of other submodules, indicating that the MRP submodules observed in the less dense fractions were not dissociation products of mtSSUs or 55S mitoribosomes caused by the experimental procedure (Extended Data Fig. 3e,f). Lastly, we generated

an mS40-knockout cell line to determine the interdependency of B5 members. Although all MRPs of B5 can still form a complex in the absence of mS40, mS40 licensed B5 for further progression into the mtSSU (Extended Data Fig. 3d).

Assembly of the HB2 module starts with association of the mtSSU platform MRPs bS1m, uS2m, uS6m and mS23 to form the B1 module, followed by engagement of uS9m during the second phase of assembly, resulting in the formation of HB1 (Figs. 2 and 3, Extended Data Fig. 2 and Extended Data Fig. 3e,f). Incorporation of the head module H1, composed of uS10m and mS35, then accomplishes the HB2 formation.

Formation of the HB4 module proceeds by association of H2 and H3 with the 12S mt-rRNA, as estimated according to the similar abundances and fluxes of the constituents (Figs. 2 and 3 and Extended Data Fig. 2). Further incorporation of the RNA-binding MRP uS14m anchors the generally poorly RNA-binding H2 cluster into the structure, while binding of uS15m might initiate the folding of the rRNA central domain as observed in bacteria[25]. Recent structural approaches revealed an alternative path for uS14m incorporation, where it associates with late-maturing mtSSU particles downstream of the incorporation of bS21m and uS11m (ref. 7).

Although the overall architecture of the mtSSU body domain is reminiscent of its bacterial counterpart, the evolution of the mtSSU, which is characterized by a substantial reduction in the rRNA content, was accompanied by the loss of the 5′ rRNA primary binding proteins uS4, uS8 and bS20 (refs. 1,2). Thus, the HB3 module structure solely relies on protein–protein interactions serving as a base for HB4 docking and subsequent 12S rRNA folding and modification.

The structure of the resulting premature mtSSU particle is further compacted by the incorporation of uS5m, uS12m, bS18m, mS33 and mS38 (Fig. 3). Correct positioning of the 3′ end of the 12S mt-rRNA is

necessary for the biogenesis of the decoding center[7]. Initial 3′ domain folding is ensured by the guanosine triphosphatase (GTPase) ERAL1 in proximity of the uS7m binding site. While uS7m depletion does not interfere with the formation of the other assembly modules, including the related H3 cluster, it prevents their association with the mtSSU particle (Extended Data Fig. 3g). The release of ERAL1 is coupled with the incorporation of the late-binding proteins uS11m and bS21m and the biogenesis factor mtRBFA (ref. 7). Additional assembly factors, crucial for late steps during mtSSU maturation, include MCAT, the methyltransferases TFB1M, METTL15 and METLL17, the GTPase MTG3 and the initiation factor mtIF3 (refs. 6,7), most of which correlate in their abundances with late-maturing mtSSU particles (Supplementary Fig. 6). The dissociation of mtRBFA exposes the binding site of mS37, the last MRP that joins the maturing particle[6].

## RNA-independent cluster formation

Remarkably, no mt-rRNA was detected in less dense fractions where assembly intermediates such as B5 or HB1 migrated, suggesting the formation of protein-only modules (Fig. 4a). To monitor the interdependency and stability of the mt-rRNAs and MRPs over time, we blocked mitochondrial transcription by treating cells with ethidium bromide[26,27]. Subsequent analysis of mt-rRNA and MRP levels revealed that constituents of these protein-only submodules, such as bS1m ($t_{1/2}$ = 15.0 h) or mS22m ($t_{1/2}$ = 12.1 h), had longer half-lives than the 12S mt-rRNA ($t_{1/2}$ = 3.5 h), indicating an RNA-independent cluster assembly (Fig. 4b and Supplementary Fig. 7). By contrast, late-binding MRPs such as uS15m ($t_{1/2}$ = 2.4 h) or uS5m ($t_{1/2}$ = 3.3 h), which depend on the presence of mt-rRNA for formation, showed a similar turnover to the 12S mt-rRNA. Proteins of one submodule, such as bS16m, uS17m, mS22, mS25, mS27 and mS34 of the B5 module, still comigrated in less dense gradient fractions when the mtSSU was absent in the ethidium bromide-treated samples, which supports the conclusion that clusters are formed independent of the 12S mt-rRNA (Supplementary Fig. 8). To dissect the nature of these protein-only modules in detail, we performed immunoprecipitation experiments upon ethidium bromide-mediated mt-rRNA depletion and purified ribosome complexes using FLAG-tagged components of these submodules (Fig. 4c). Using bS1m[FLAG] as a bait, all tested MRPs were coimmunoprecipitated, indicating the purification of mtSSUs and 55S mitoribosomes in the untreated sample, whereas the majority of MRPs were not detectable in the elution upon mt-rRNA depletion (Fig. 4d). However, constituents of the B1 and HB1 complexes were efficiently copurified with bS1m[FLAG] in the absence of mt-rRNA, confirming the RNA-independent formation of these submodules. Similarly, components of the B5 complexes were coprecipitated using mS27[FLAG] independent of mt-rRNA availability (Fig. 4e). To further confirm the stable assembly of the B5 complex in mt-rRNA-depleted cells, mS22-copurified complexes were separated by high-resolution sucrose gradient centrifugation (Fig. 4f). Indeed, all investigated constituents comigrated in less dense fractions, while the complete 28S and 55S particles were not detectable in RNA-ablated cells. Taken together, assembly of the mtSSU is achieved by the formation of stable protein-only submodules, which are available in excess and remain stable in the absence of mt-rRNA, suggesting that protein–protein interactions are far more important for ribosome assembly in human mitochondria than in bacteria.

## Kinetics of the mtSSU assembly pathway

Having reconstructed the mtSSU assembly pathway, we next used mathematical modeling to estimate the impact of assembly steps on mtSSU abundance. We derived reactions for each MRP and the corresponding mtSSU modules with kinetic rates that describe the mtSSU assembly steps (Extended Data Fig. 4a). The binding and unbinding rates characterize the association and dissociation of MRPs and modules, while the MRP supply rates and turnover rates define the transport of MRPs into mitochondria and their recycling, respectively.

Kinetic rates were informed by the pulse–chase triple-SILAC experimental MS data using Bayesian inference[28] (Extended Data Fig. 4a and Supplementary Fig. 9). The constructed mathematical model was able to explain the experimental data (Supplementary Fig. 10), allowing us to obtain estimates of all kinetic rates (Supplementary Fig. 11 and Supplementary Table 6). Subsequent local sensitivity analysis (Methods, 'Local sensitivity analysis') allowed the determination of nonsensitive kinetic rates, the change of which had little to no impact on the mtSSU abundance, as well as sensitive kinetic rates (enhancing and inhibiting), the change of which had a strong impact on mtSSU abundance (Extended Data Fig. 4a). mtSSU biogenesis appears to be robust to the changes in cellular homeostasis, as alterations in the majority of kinetic rates did not interfere with the mtSSU steady-state abundance (Extended Data Fig. 4b–d). However, the kinetic rates of the group of late-binding MRPs (uS5m, uS11m, uS12m, bS18m, bS21m, mS33, mS37 and mS38) and the MRPs involved in HB4 formation represent an exception of this general trend as their increased supply and binding rates boosted mtSSU abundance up to threefold. Moreover, the abundance of mtSSUs was highly sensitive to changes in the turnover rates and unbinding rates of the same group of MRPs. Therefore, mtSSU abundance is most sensitive to late assembly steps involving the 12S mt-rRNA-associated module HB4, whereas earlier assembly steps involving protein-only assembly modules appear to have less impact on mtSSU abundance.

## The assembly pathway of the mtLSU

The formation of the highly conserved peptidyltransferase center of the mtLSU represents the terminal maturation step and requires several assembly factors such as GTPBP5, GTPBP6, GTPBP7, GTPBP10, MRM2, MRM3, DDX28, MTERF4–NSUN4 or the MALSU1 module[8–14], the relative abundances of which correlate with MRPs present in gradient fractions 8 and 9 (Supplementary Fig. 6). The earliest structurally resolved intermediates of the mtLSU included almost all MRPs, with just bL33m, bL35m and bL36m lacking, but showed a largely immature interface with unfolded rRNA[12]. We focused on upstream events in the mtLSU assembly pathway, elucidating the composition of mitoribosome assembly intermediates and the mechanism of their formation in vivo (Fig. 5 and Extended Data Figs. 5 and 6).

On the basis of their similar abundances and fluxes among the gradient fractions, we categorized the mtLSU MRPs into five major assembly clusters (Extended Data Figs. 5 and 6 and Supplementary Fig. 12). Fitting these MRP groups to the structure of the mature mtLSU, the modules were predominantly clustered along with its key architectural features: the central protuberance (CP), the L7/L12 stalk (ST), the polypeptide exit tunnel (PET), the mitoribosome anchor module (A), and the central body module (BM).

### The CP module

Otherwise conserved structural elements of the CP, such as 5S rRNA, uL5 and bL25, were lost during evolution of the human mtLSU[29]. Assembly of the CP involves the formation of two major MRP clusters, which associate with mt-tRNA[Val] (Fig. 5 and Extended Data Figs. 5 and 6). The first module (CP5) consists of uL18m and mL38 and partly constitutes the CP platform. The second subassembly (CP4) contains the mitochondrion-specific proteins mL40, mL46 and mL48, as well as other constituents of the platform including mL62 and bL31m. Formation of the two major clusters CP4 and CP5 is independent as we observed different patterns of their respective fluxes and abundances across fractions for these modules (Extended Data Fig. 6). Immunoisolation assays followed by native complex separation using high-resolution sucrose gradient sedimentation did not reveal an association of the mL62-containing and bL31m-containing complex with uL18m or mL38 during early subassembly (Extended Data Fig. 7a,b). Importantly, we did not detect any other modules, including the BM or A modules, in less dense fractions when separating mL62-containing or bL31m-containing complexes; vice versa, CP4 or CP5 were not observed

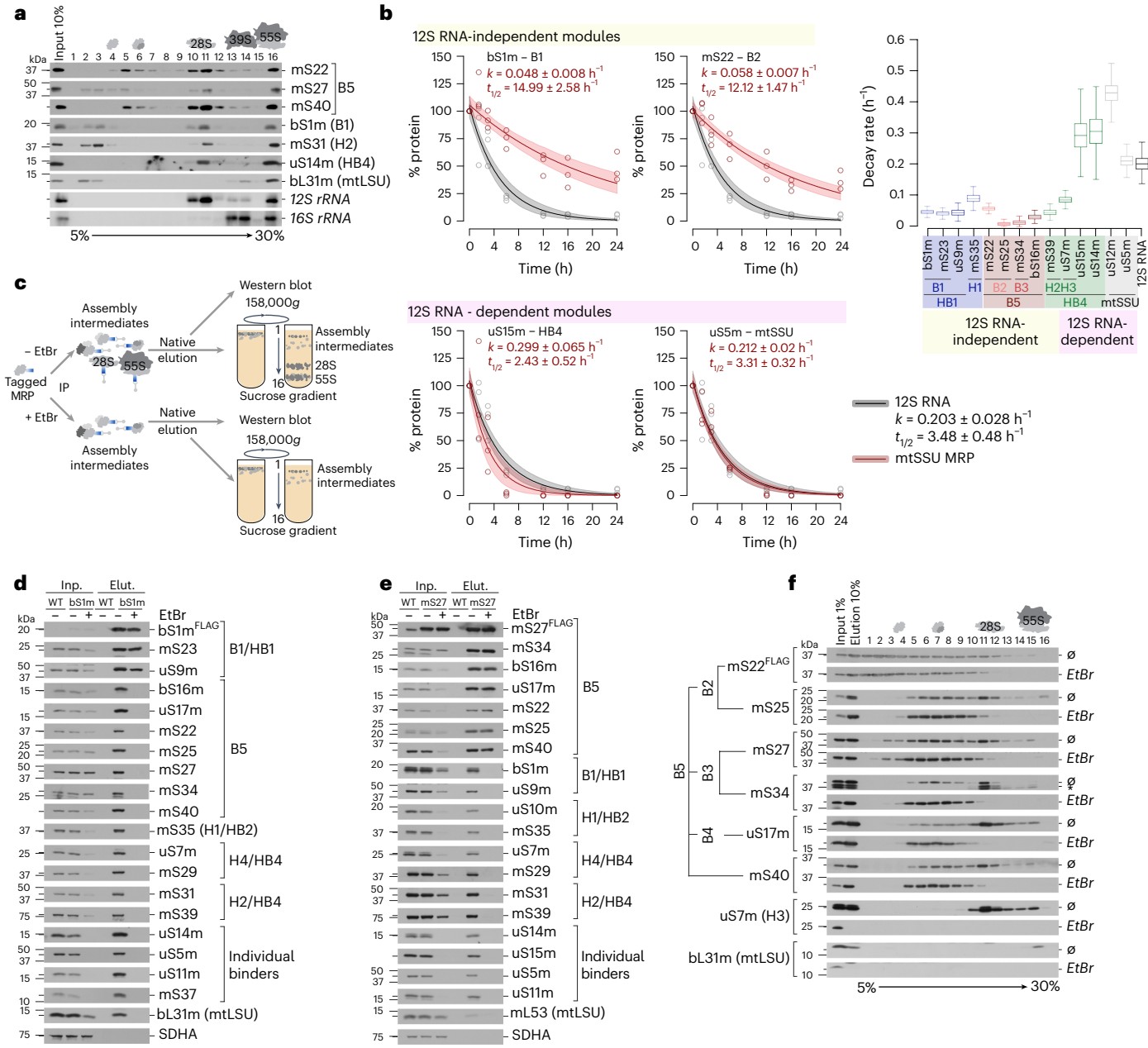

**Fig. 4 | Formation of assembly modules is independent of the presence of rRNA. a**, rRNA and MRP distribution across sucrose gradient fractions. Mitoribosome complexes were isolated from HEK293 wild-type cells and separated by sucrose gradient ultracentrifugation (high-resolution gradient, 158,000*g*, 15 h). MRP distribution across fractions was detected by western blotting with indicated antibodies. RNA was isolated from collected fractions and analyzed by northern blotting using probes against *mtRNR1* (12S rRNA) and *mtRNR2* (16S rRNA). **b**, MRP turnover upon repression of mt-rRNA synthesis by ethidium bromide (EtBr). Plotted are the relative MRP (red) and 12S rRNA (gray) abundance at indicated time points after treatment as a percentage of the starting abundance (time point, 0 h). Solid lines indicate the median and shaded

areas indicate the 5th and 95th percentiles of model fits using *n* = 3 biological replicates. Right, the decay of 12S rRNA-independent versus 12S rRNA-dependent assembly modules or individual MRPs (Supplementary Fig. 7). $t_{1/2}$, half-life; *k*, decay rate. Box plots indicate the median, first quartile, third quartile and minimum and maximum after outlier removal. **c**, Experimental setup for validation of rRNA-independent nature of MRP assembly modules. **d,e**, Immunoisolation of assembly modules using FLAG-tagged constituents bS1m (**d**) and mS27 (**e**) in the absence of rRNA (with EtBr). **f**, Formation of the B5 assembly module in the absence of rRNA. B5 was isolated using FLAG-tagged mS22 in the presence (Ø) or absence (with EtBr) of 12S rRNA and separated by sucrose gradient ultracentrifugation.

in less dense fractions when isolating mitoribosome complexes using components of other modules such as mL44^FLAG of the BM (Extended Data Fig. 9a). Thus, we can exclude that these subassemblies were dissociation products of the mtLSU arising because of the experimental progress. Assembly of the CP4 and CP5 modules does not require the presence of mt-tRNA^Val as the complex persisted even upon mt-RNA depletion (Extended Data Fig. 7a,c). The mt-RNA-independent nature

of the CP assembly complex was previously suggested in yeast[30] and in human cells lacking mtDNA[31]. CP attachment finalizes formation of the basic mtLSU architecture as we observed the presence of pre-mtLSU particles lacking the CP that migrated in fraction 7 in mL62-deficient cells (Extended Data Fig. 7d). This is in line with the bacterial LSU and yeast mtLSU biogenesis where CP integration is one of the final events during LSU assembly[30,32,33]. The formation of mitoribosomes

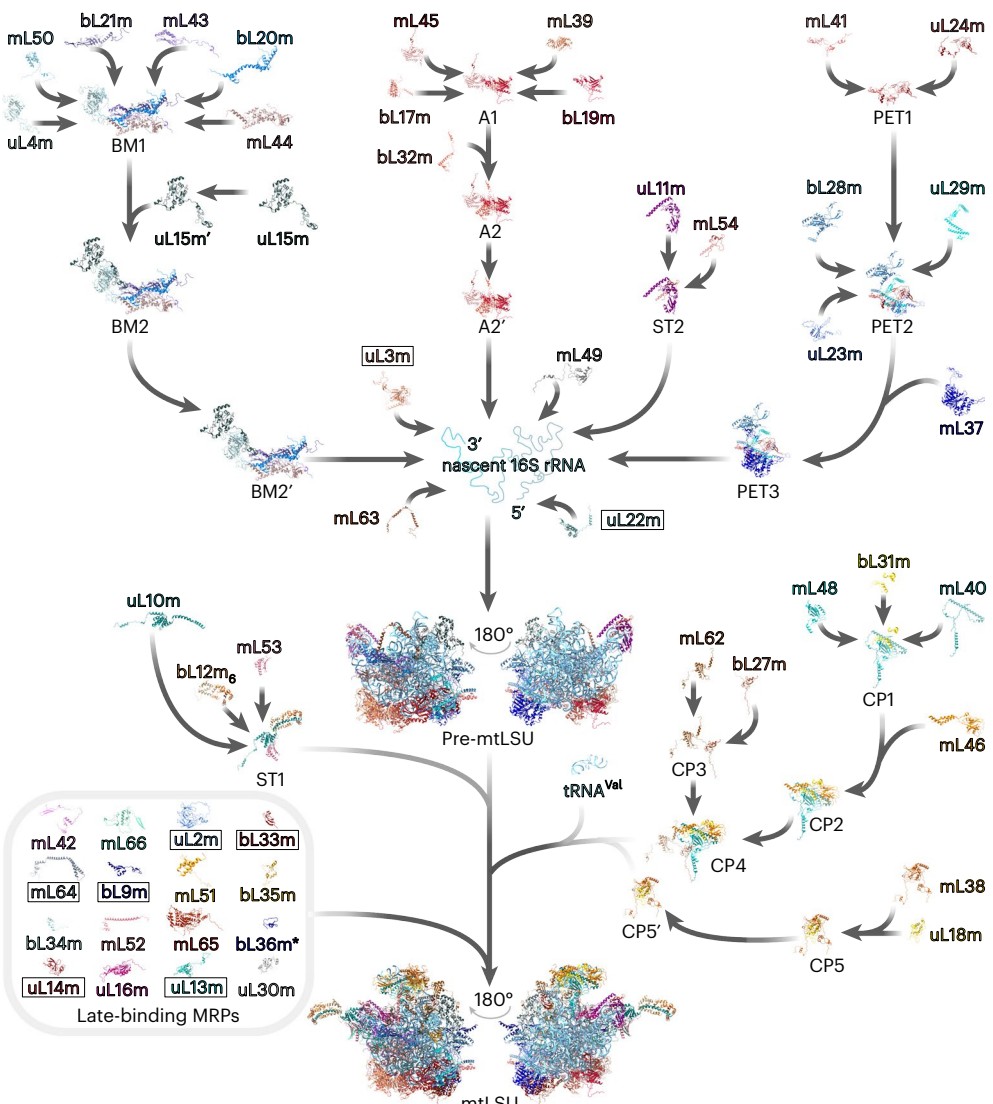

**Fig. 5 | Reconstructed in vivo pathway of the mtLSU assembly.** The biogenesis of the 39S mtLSU entails a stepwise association of preassembled MRP clusters with the 16S rRNA elements. uL22m and uL3m bind the 5′ and 3′ ends of the 16S rRNA, respectively, to launch domain compaction and mtLSU assembly. During the late stages of assembly, a set of MRPs attach to the maturing mtLSU as discrete proteins, refining already established biogenesis intermediates. The prime symbol indicates differences in the kinetic properties of the MRPs and assembly modules with the identical MRP composition. Boxed MRPs were detected in the low-density gradient fraction but not involved in the MRP assembly modules. *bL36m was placed as a late-assembling protein according to previous studies[12,14] but was not continuously detected during our MS analysis. uL1m is not included in the assembly scheme as it was not resolved in the mitoribosome structure used for the pathway reconstruction (PDB 6ZM6).

was notably impaired in mL62-ablated cells; however, mitoribosomal particles lacking mL62 retained residual translation activity, indicating that the mtLSU can assemble following alternative paths, although the efficiency is drastically reduced (Extended Data Fig. 7e).

### The ST module

The ST assembly cluster consists of two MRP groups, which are incorporated into the maturing mtLSU particle at different stages (Fig. 5 and Extended Data Figs. 5 and 6). The first subassembly (ST2) contains uL11m and mL54, the two MRPs forming the mitoribosomal stalk base (Extended Data Fig. 8a). The second module (ST1) includes six copies of bL12m, which are organized around uL10m and additionally stabilized by mL53 (ref. 34) (Extended Data Fig. 8b,c). The presence of a large pool of free bL12m (Fig. 1c and Extended Data Fig. 5) reflects its secondary function as it stabilizes POLRMT (mtDNA-directed RNA polymerase) in vivo and serves as an activator of mitochondrial transcription in vitro[35,36]. Accordingly, we detected bL12m in association

with POLRMT but this was independent of ST1 formation and did not involve uL10m (Extended Data Fig. 8b,c). Although ST1 joins the maturing mtLSU at a late stage, its engagement precedes CP integration as we detected uL10m and bL12m in particles lacking the CP (Extended Data Fig. 7d). Depletion of mt-rRNA does not affect formation of the ST module as revealed by coimmunoprecipitation experiments using bL12m^FLAG or uL11m^FLAG as baits (Extended Data Fig. 8d,e) and also supported by the accumulation of ST1 and ST2 clusters in the low-density gradient fractions (Extended Data Fig. 8f).

### The BM

Although the BM includes additional mitochondrion-specific proteins, its core is conserved from bacteria[37–39]. The BM comprises conserved primary mt-rRNA-binding proteins uL4m and bL20m (Fig. 5 and Extended Data Figs. 5, 6 and 9a), the bacterial homologs of which bind proximal to the 5′ end of the 23S rRNA during early assembly[37,39]. However, mitochondrial BM maturation involves a complex network

of protein–protein interactions independent of the 16S mt-rRNA (Extended Data Fig. 9b,c). Ablation of bL20m completely abolished bL21m and mL43 incorporation, while the other constituents of the module, although greatly reduced, formed a smaller complex migrating in low-density gradient fractions (Extended Data Fig. 9d). The absence of uL4m prevented mL50 recruitment to the BM, while the other constituents were able to form a subassembly. In bacteria, uL4 facilitates folding of the 23S rRNA domain II, enabling the recruitment of intermediate binding proteins[40]. These differences highlight the divergence of these two assembly pathways, further supporting the formation of mt-rRNA-independent preassembly units. The presence of the disease-associated protein mL44 (refs. [41],[42]) is essential for mL43 recruitment. A lack of mature BM because of the ablation of its constituents does not affect the formation of the other assembly clusters of the mtLSU, emphasizing the independent formation of mitoribosome modules before cluster joining (Extended Data Fig. 9d). The kinetic behavior of uL15m suggests that uL15m is a late-binding protein (Extended Data Fig. 5 and Supplementary Data 3 and 7). However, on the basis of its cosedimentation profile with mL44[FLAG]-containing complexes (Extended Data Fig. 9a) and its detection in the less dense gradient fraction in perturbation experiments (Extended Data Fig. 9c,d), we placed uL15m in BM2 but we cannot exclude that it is rather a late-binding protein.

### The PET and A modules

The mitochondrial translation apparatus has been evolutionary adapted for the synthesis of hydrophobic membrane proteins, apparent in its membrane association as a function of the A module and the hydrophobic nature of the PET[43–46] (Fig. 5 and Extended Data Figs. 5 and 6). Human mitoribosomes are associated with the inner mitochondrial membrane by mL45, which forms, with bL17m, bL19m and mL39, a subassembly of the A module that is further supplied with bL32m (Extended Data Fig. 10a). Similar to the other mtLSU MRP clusters, the A module formation is independent of mt-rRNA (Extended Data Fig. 10b). The A module and, thus, membrane association are essential for mtLSU biogenesis as no mtLSU particles were formed in the absence of mL45 (Extended Data Fig. 10c). The human mitoribosomal PET consists of the bacterial homologs uL22m, uL23m, uL24m and uL29m (refs. [47],[48]). These proteins, except uL22m, form the PET assembly module together with the mitochondrion-specific MRPs mL37 and mL41 (Fig. 5 and Extended Data Figs. 5, 6 and 10d). The flux of uL22m did not correlate with the PET or A clusters although, in the mature mtLSU, the MRP is enclosed by mL39, mL45 and uL23m (Supplementary Data 7 and 9). It is tempting to speculate that uL22m is primed for association with the 16S mt-rRNA by one or more assembly factors, as we detected uL22m-containing complexes in the low-density gradient fractions (Extended Data Fig. 10e). Remarkably, assembly of the bacterial LSU is initiated by engagement of uL22, uL24 and uL29 with domain I of the 23S rRNA, which further serves as a platform for folding of the other domains[40]. Thus, uL22m could initiate the 5′ rRNA folding of the mtLSU, similar to its bacterial homolog.

### Assembly of pre-mtLSU and mtLSU

A group of MRPs including uL22m and uL3m, which all form extensive interactions with the 16S mt-rRNA, associate as individual proteins (Fig. 5, Extended Data Figs. 5 and 10e and Supplementary Fig. 4b). The binding sites of these MRPs span multiple 16S mt-rRNA domains in the mature mtLSU[47,48]. Thus, their joining reduces the conformational freedom of the 16S mt-rRNA and, together with the binding of mtLSU assembly modules, promotes mt-rRNA folding. Interestingly, assembly of bacterial LSUs starts with base pairing between the 5′ and 3′ ends[49]. Although the 5′ and 3′ ends of the 16S mt-rRNA are in proximity in the mature mtLSU, they do not base pair[47,48]. Contrary to the bacterial system, the bridge between the ends is mediated by proteins highlighting the evolutionary divergence. While uL3m binds to the 3′ end of the 16S

mt-rRNA, uL22m contacts the 5′ end and bL32m of the A module forms a bridge between both proteins, thus connecting the 16S mt-rRNA ends. Afterward, other MRPs and ST1 join the maturing mtLSU particle and CP installation finalizes formation of the mtLSU (Fig. 5 and Extended Data Fig. 7d). In the mature particle, the CP is bridged to the main body by mL52 and mL64 (refs. [47,48]), suggesting that these proteins might associate with the mtLSU afterward to anchor the CP module. Maturation of the interfacial mt-rRNA, which is mediated by auxiliary factors, such as DDX28, MRM3 and GTPBP10, further facilitates the incorporation of bL33m, bL35m and bL36m (refs. [12,14]). The catalytic core of the mtLSU is finally matured by multiple assembly factors, including GTPBP5, GTPBP6, GTPBP7, MRM2 and MTERF4–NSUN4 (refs. [8–14]). According to the flux analysis, the incorporation of uL1m into the 16S mt-rRNA moieties that scaffold the mtLSU L1 stalk occurs after the pre-mtLSU formation (Supplementary Data 7), probably after the incorporation of bL9m, the bacterial homolog of which stabilizes the stalk base[40]. We did not include uL1m in the final mtLSU assembly model as it is absent from the structure used for the pathway reconstitution (Protein Data Bank (PDB) 6ZM6).

In principle, the assembly modules span around multiple rRNA domains, regardless of the 5′-to-3′ direction of the transcription. It is unlikely, therefore, that much of the mitoribosome assembly occurs cotranscriptionally. Moreover, occlusion of nascent mt-rRNA by large MRP subassemblies would interfere with rRNA processing, which is crucial for mitoribosome biogenesis[23].

## Discussion

Our data suggest a fundamentally different pathway for the assembly of human mitoribosomes compared to their bacterial or cytosolic counterparts. The initial stages of biogenesis involve intensive interactions of MRPs to form proteinaceous subassemblies that migrate in less dense gradient fractions, in which mt-rRNA is not detectable (Fig. 4)[23,50,51]. These protein-only modules are formed in the absence of mt-rRNA, which is in agreement with the higher stability of MRPs in comparison to mt-rRNAs. It remains to be addressed whether assembly factors are required for the formation of these protein-only modules and how their association influences mt-rRNA folding dynamics; however, this study provides a fundamental basis for future studies focusing on this aspect. Additionally, the presence of late-binding proteins such as uS11m or uS12m in less dense fractions suggests the formation of complexes potentially involving nonribosomal entities. As assembly proceeds, the preformed complexes and individual RNA-binding proteins associate with mt-rRNA, contributing to its folding. The nearly mature structure is consolidated by the late-binding proteins, which serve as molecular clips. The formation of robust protein-only subassemblies during both mtSSU and mtLSU biogenesis suggests that rRNA synthesis is likely a rate-limiting step during mitoribosome assembly. After mt-rRNA becomes available, subunit construction can start immediately through the incorporation of pre-existing protein modules. The incorporation of preassembled MRP complexes during mitoribosome biogenesis was previously suggested in yeast and trypanosoma[30,52] and reflects a distinct assembly pathway of protein-rich mitoribosomes, which have to deal with supernumerary proteins.

## Online content

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

[1]Department of Molecular Biology, University Medical Center Göttingen, Göttingen, Germany. [2]Department of Cellular Biochemistry, University Medical Center Göttingen, Göttingen, Germany. [3]Cluster of Excellence 'Multiscale Bioimaging: from Molecular Machines to Networks of Excitable Cells' (MBExC), University of Göttingen, Göttingen, Germany. [4]Bioanalytical Mass Spectrometry Group, Max Planck Institute for Multidisciplinary Sciences, Göttingen, Germany. [5]Quantitative and Systems Biology Group, Max Planck Institute for Multidisciplinary Sciences, Göttingen, Germany. [6]Centre for Inflammation Biology and Cancer Immunology & Peter Gorer Department of Immunobiology, King's College London, London, UK. [7]Francis Crick Institute, London, UK. [8]Bioanalytics, Institute for Clinical Chemistry, University Medical Center Göttingen, Göttingen, Germany. [9]Göttingen Center for Molecular Biosciences, University of Göttingen, Göttingen, Germany. [10]These authors contributed equally: Elena Lavdovskaia, Elisa Hanitsch, Andreas Linden. ✉e-mail: henning.urlaub@mpinat.mpg.de; juliane.liepe@mpinat.mpg.de; ricarda.richter@med.uni-goettingen.de

## Methods

### Cell culture and stable isotope labeling

HEK293 cells (human embryonic kidney cells) were incubated under standard culture conditions at 37 °C in a humidified atmosphere enriched with 5% $CO_2$ in DMEM high-glucose medium (Capricorn Scientific) with 10% dialyzed FBS. For SILAC, cells were cultured in DMEM for SILAC (Thermo Fisher) containing 10% dialyzed FBS, devoid of Arg and Lys and supplemented with either 84 mg $l^{-1}$ Arg-10 and 146 mg $l^{-1}$ Lys-8 (H SILAC medium) or the same concentrations of Arg-6 and Lys-4 (M SILAC medium) (Cambridge Isotope Laboratories). Before the experiment, cells were treated with H SILAC medium for 10 days. To monitor protein turnover in mitoribosome assembly intermediates, cells were briefly rinsed with PBS pH 7.4 (137 mM NaCl, 2.7 mM KCl, 10 mM $Na_2HPO_4$ and 1.8 mM $KH_2PO_4$) and then chased on M SILAC medium for 1.5, 3, 6 or 12 h or used without a chase (0 h). To assess global protein turnover in the total cell or mitochondrial fraction, cells were first chased for 12 h in M SILAC and collected immediately (0 h) or additionally chased in standard DMEM containing L isotopes for 3, 6, 9, 12, 15, 18, 21 and 24 h. After harvesting, the cell pellets were stored at −80 °C until further processing.

### Mitochondrial isolation from cultured cells and mitoplasts preparation

Mitochondria and mitoplasts were isolated as described previously[8]. Briefly, cells were homogenized in trehalose buffer (300 mM trehalose, 10 mM KCl, 10 mM HEPES–KOH pH 7.4, 1 mM PMSF and 0.2% (w/v) BSA) using a Homogenplus homogenizer (Schuett-Biotec). After centrifugation of the cell homogenate at 400$g$ for 10 min at 4 °C, mitochondria were recovered at 11,000$g$ for 10 min at 4 °C and frozen at −80 °C, treated with proteinase K (1:200 ratio of proteinase K to mitochondria) to obtain mitoplasts or used immediately for further applications.

### Sucrose gradient ultracentrifugation

Isolated mitochondria or mitoplasts (550 μg) were lysed in lysis buffer (3% (w/v) sucrose, 100 mM $NH_4Cl$, 15 mM $MgCl_2$, 20 mM Tris-HCl pH 7.5, 1% (w/v) digitonin, 0.08 U per μl RiboLock RNase Inhibitor (Thermo Fisher) and cOmplete protease inhibitor cocktail (Roche)) for 30 min at 4 °C with gentle shaking and the resulting lysate was cleared at 16,000$g$ for 15 min at 4 °C before loading onto a sucrose gradient (5–30% (w/v) sucrose, 100 mM $NH_4Cl$, 10 mM $MgCl_2$, 20 mM Tris-HCl pH 7.5 and cOmplete protease inhibitor cocktail (Roche)). The mitoribosomal complexes were separated by ultracentrifugation for 15 h at 79,000$g$ (21,500 r.p.m.; low-resolution gradient) or for 15 h at 158,000$g$ (30,400 r.p.m.; high-resolution gradient) using an SW41Ti rotor (Beckman Coulter). Gradient fractions (1–16) were collected using a BioComp gradient station from top to bottom and the proteins were precipitated from solution with 2.5 volumes of 95% ice-cold ethanol and 0.1 volumes of 3 M sodium acetate pH 5.2. For liquid chromatography (LC)–MS/MS analysis, an internal standard of unlabeled L mitoribosomes was added to each fraction of the three biological replicates.

### Preparation of the mitoribosome standard

Mitoribosomes were isolated following established protocol[8]. Mitoplasts were disrupted in a lysis buffer (20 mM Tris-HCl pH 7.4, 100 mM $NH_4Cl$, 15 mM $MgCl_2$, 2 mM DTT and 1% Triton X-100). After centrifugation at 16,000$g$ for 15 min at 4 °C, the resulting lysate was layered onto a two-step sucrose cushion (1 M–1.75 M sucrose cushion) and centrifuged for 15 h at 148,000$g$ at 4 °C. Fractions were collected from top to bottom of the cushion and the fraction containing mitoribosome particles was concentrated and subjected to buffer exchange with wash buffer (100 mM $NH_4Cl$, 15 mM $MgCl_2$, 20 mM Tris-HCl pH 7.4 and 2 mM DTT). The concentrated sample was layered onto a 15–30% sucrose gradient (15–30% w/v sucrose, 100 mM $NH_4Cl$, 15 mM $MgCl_2$ and 20 mM Tris-HCl pH 7.4) and centrifuged (115,600$g$ for 16 h and 10 min at 4 °C) using an SW41Ti rotor (Beckman Coulter). Fractions containing

55S mitoribosomes were further concentrated and washed with wash buffer. The composition of the final sample was confirmed by western blot and label-free MS.

### Sample preparation, LC–MS analysis and database search

For each pulse–chase SILAC experiment (that is, using total cell lysate and isolated mitochondria), three biological replicates and two technical replicates from each biological replicate were analyzed. Samples were lysed in lysis buffer (50 mM Tris-HCl pH 7.4, 130 mM NaCl, 2 mM $MgCl_2$, 1% NP-40, 1 mM PMSF and 1× cOmplete protease inhibitor cocktail (Roche)) and adjusted with NuPAGE sample buffer (ThermoFisher Scientific). Proteins from the different chase time points were separated by SDS gel electrophoresis using precast NuPAGE Bis-Tris gels (ThermoFisher Scientific). For the analysis of cell lysate, the samples were separated using the entire molecular weight range of the gel and each lane representing each time point was cut into 23 gel slices; proteins were reduced, alkylated and digested in gel with trypsin overnight. Peptides were extracted, dried in a SpeedVac, resuspended in 2% (v/v) acetonitrile with 0.05% (v/v) trifluoroacetic acid and analyzed by LC–MS/MS. For the analysis of mitoribosomal complexes, the pelleted proteins from gradient fractions of each chase time point were dissolved in NuPAGE sample buffer (ThermoFisher Scientific) and proteins were separated by NuPAGE gels (ThermoFisher Scientific) in such a manner that the samples were allowed to run only half of the gel size. The lanes were cut into four slices and processed as described above.

LC–MS/MS analysis was performed in the same manner for peptides derived from digested cell lysate and from sucrose density gradient centrifugation. Peptides were analyzed on a QExactive HF MS instrument coupled to a Dionex UltiMate 3000 UHPLC system (both Thermo Fisher Scientific) equipped with an in-house packed C18 column (ReproSil-Pur 120 C18-AQ; pore size, 1.9 μm; inner diameter, 75 μm; length, 30 cm; Dr. Maisch). Acquisition was controlled and monitored using Thermo Xcalibur Instrument Setup (version 4.4.16.14) and Tune Application (version 4.0.309.28). Peptides were separated applying the following gradient: mobile phase A, 0.1% (v/v) formic acid; mobile phase B, 80% (v/v) acetonitrile with 0.08% (v/v) formic acid. The gradient started at 5% B, increasing to 10% B within 3 min, followed by a linear increase to 46% B within 45 min and then keeping B constant at 90% for 8 min. After each gradient, the column was again equilibrated to 5% B for 2 min. The flow rate was set to 300 nl $min^{-1}$. MS1 full scans were acquired with a resolution of 60,000, a maximum injection time (IT) of 50 ms and an automatic gain control (AGC) target of $1 \times 10^6$. Dynamic exclusion was set to 30 s. MS2 spectra were acquired for the 30 most abundant precursor ions; the resolution was set to 15,000, the maximum IT was set to 60 ms and the AGC target was set to $1 \times 10^5$. Fragmentation was enforced by higher-energy collisional dissociation at 28% normal collision energy. Acquired raw data were searched against a reviewed (Swiss-Prot) human reference proteome database, downloaded from UniProt Knowledgebase, using MaxQuant software[53] (version 1.6.0.1) applying default settings with the following exceptions: multiplicity, 3 (M, Arg-6 and Lys-4; H, Arg-10 and Lys-8); matching between runs, enabled; fixed modification, carbamidomethylation; variable modifications (included in protein quantification), oxidation (M) acetylation (protein N terminus); enzyme 'Trypsin/P'; enzyme mode, 'specific'; maximum missed cleavages, 2. Precursor and MS2 mass tolerance were set to 4.5 ppm and 20 ppm, respectively.

### MS data analysis

Data analysis for the next section was performed in Python. All remaining data processing, modeling and subsequent model downstream analysis was performed in R version 4.1.0 using the packages 'vroom_1.6.1', 'dplyr_1.0.9', 'stringr_1.4.0' and 'tidyr_1.1.4' or Python (version 3.9.17), if not further detailed below[54–58].

**Protein turnover estimation—no sucrose gradient.** Whole-cell lysate and isolated mitochondria samples derived from the triple-SILAC pulse–chase experiment measured by MS were processed to determine the turnover of MRPs and ribosomal assembly factors. Protein turnover in cell culture experiments is commonly modeled as exponential decay (ED)[19,20]. However, the turnover of proteins that undergo stable complex formation was described as nonexponential decay (NED), whereby free subunits are turned over faster compared to subunits that are incorporated into their complex[21]. In such a scenario, the protein of interest exists in two states, A and B, where state A describes the free subunits and state B describes the protein complex. To derive the turnover rates of MRPs and their assembly factors, we adapted the ED and NED models described in the literature[21] by extending them to allow nonconstant cell growth over the time course of the full pulse–chase experiment, resulting in a one-state-model and a two-state-model.

In the one-state-model (only state A) proteins are produced with rate $p$ and degraded with rate $k_a$. In the two-state-model (state A and state B), proteins also transfer from state $A$ to state $B$ with rate $k_{ab}$ and proteins in state $B$ are degraded with rate $k_b$ (Extended Data Fig. 1b). The one-state-model can be described by the following ordinary differential equation (ODE):

$$\frac{d\,A(t)}{d\,t} = p \times C(t) - k_a \times A(t).$$

The two-state-model is defined by the following set of ODEs:

$$\frac{d\,A(t)}{d\,t} = p \times C(t) - (k_a + k_{ab}) \times A(t),$$

$$\frac{d\,B(t)}{d\,t} = k_{ab} \times A(t) - k_b \times B(t),$$

with

$$C(t) = \begin{cases} e^{g_m \times (t+12)}, & t < 0 \\ e^{g_m \times 12 + g_l \times t}, & t \geq 0 \end{cases}$$

and

$$g_m = \frac{\log(2)}{t_m}; g_l = \frac{\log(2)}{t_l}.$$

$A(t)$ is the amount of protein in state A at time $t$, $B(t)$ is the amount of protein in state B at time $t$, $C(t)$ is the number of cells at time $t$, $p$ is the production rate (defined by $k_a$, $k_{ab}$, $k_b$ and $g_1$ or $g_2$), $k_a$ is the degradation rate of state A, $k_b$ is the degradation rate of state B, $k_{ab}$ is the transfer rate from state A to state B, $t_m$ is the doubling time of cells in the M chase (time it takes cells to double in number) and $t_l$ is the doubling time in the L chase. To note, the one-state model is equivalent to the two-state-model with $k_{ab} = 0$ and B(0) = 0. Therefore, a likelihood-ratio test for nested models can be used to determine the model that best describes the data, as elaborated on below.

To determine the cell doubling times, we selected a set of unrelated proteins (that is, proteins that were not MRPs or their assembly factors) detected in the MS data. MaxQuant search results file 'proteins.txt' was exported to extract MRP intensity values for H, M and L labeled proteins over time (0–24 h) for both isolated mitochondria and whole-cell lysate. H, M and L intensity values were divided by their sum to represent proportions. Time points 0 and 24 h were excluded from data analysis because they were incoherent with the remaining time points (Supplementary Data 1) (for example, the M signal at 0 h was repeatedly smaller than the M signal at 3 h or 6 h, while the L signal at 24 h was repeatedly smaller than the L signal at 21 h or 18 h), likely because of experimental error.

Unrelated proteins were filtered by variance between replicates (variance < 0.9 quantile) by proportion of the H isotope at 18 h (0.925 quantile < proportion < 0.99 quantile) and 21 h (0.95 quantile < proportion < 0.99 quantile) and by complex formation (that is, proteins that did not form large and stable complexes) to obtain abundant proteins that were stable over time and most likely best described by the one-state-model (Supplementary Fig. 1a). We assumed that the decrease in H isotope in this selected stable protein pool was dominated by dilution with M and L isotopes caused by cell division and growth during the experiment but not mainly by the actual degradation of H labeled proteins. We observed that one constant growth rate over the whole experiment could not explain the data but the model with one growth rate during the M chase and one growth rate during the L chase could (Supplementary Fig. 1b). The pool of unrelated proteins was modeled with the one-state-model and two growth rates. Growth rates were treated as global parameters shared between proteins. The inferred growth rates (Supplementary Fig. 1c) were then used to infer the degradation and transfer rates of MRPs and ribosome assembly factors (Supplementary Fig. 1d).

Every isotope has its own model governed by the same ODEs but differs by the initial states and the production rate. Before the start of the first pulse ($t = -12$ h) all proteins are H labeled; hence, the proportion of H is 1 and the proportion of M and L is 0. During the M chase, the production rate of H and L is 0 and the production rate of M is $p$ (as the proportion of H decreases, the proportion of M increases and the proportion of L stays 0). During the L chase, production rate of L is $p$ while the production rate of H and M is 0 (as the proportion of H and M decreases, the proportion of L increases).

Observed data were modeled as the model's prediction plus additive error from a normal distribution centered around 0 with s.d. (inferred parameter). We used Matlab (version 9.13.) to obtain an algebraic solution of the ODE systems of the one-state-model and two-state-model. Parameter inference was realized using the no U-turn sampler (NUTS) implemented in the PyMC (version 5.7.2) Python library function PyMC.sample()[59]. Supplementary Table 7 provides a definition of prior distributions of the kinetic rates. $t_m$ and $t_l$ were treated as parameters only in the modeling of unrelated stable proteins; for modeling of proteins of interest (that is, MRPs and their assembly factors), they were set as constants derived as the median from the marginal posterior distribution of unrelated proteins ($t_m = 46.4$ h and $t_l = 30.8$ h). Model inference was performed for $10^4$ iterations. Proteins that did not converge were tuned for a further $10^5$ iterations. Convergence was automatically checked using r-hat statistics[60] and manually inspected in the end. To decide whether the one-state-model or the two-state-model better describes the data, we used the likelihood-ratio test[61] (waging the simplicity of the model against capability to explain the observed data) with $P$ values < 0.05 indicating the two-state-model to be notably better than the one-state-model.

**MS data normalization—sucrose gradient fractions.** MaxQuant search results file 'proteins.txt' was exported to extract MRP intensity values for H, M and L labeled proteins over time (0–12 h) and across sucrose gradient fractions (Supplementary Data Figs. 2 and 3). The total MRP intensity was the sum of H and M labeled MRP intensities. An L labeled standard of isolated mitoribosomes spiked into each sample was used to obtain MRP abundances (in arbitrary units relative to the standard) that were comparable across different MRPs by dividing H and M labeled MRP intensities by L labeled MRP intensities (Fig. 1a). Under steady-state conditions, the total abundance of each MRP in a given sucrose gradient fraction is constant over time. Therefore, MRP abundance values of a given sucrose gradient fraction were normalized such that their total abundance at each measurement time point was equal to the mean total MRP abundance in that sucrose gradient fraction across time. MRPs involved in mtSSU assembly were further normalized to the mtSSU abundance observed in sucrose gradient fraction

7 (Supplementary Data 4 and Supplementary Table 2). Resulting values correspond to the abundance of an MRP in a given sucrose gradient fraction relative to the mtSSU steady-state abundance. Accordingly, MRPs involved in mtLSU assembly were normalized to the mtLSU abundance observed in sucrose gradient fraction 8 (Supplementary Data 5 and Supplementary Table 2).

**Construction of mtSSU and mtLSU assembly pathway.** The obtained normalized steady-state abundances of each MRP across sucrose gradient fractions provides a first layer of information about the mtSSU assembly pathway. However, we observed that many MRPs showed similar abundances in the same fraction. This was especially the case for fractions 1 and 2. However, it is unlikely that ten or more MRPs formed modules in those early fractions. The exchange of H to M labeled MRPs across sucrose gradient fractions and over time allowed gaining further insight into the mtSSU assembly steps and the involved interaction partners. The idea is that two MRPs that are part of the same module would appear with comparable abundance in the same fraction and all proceeding fractions. Furthermore, the exchange of H labeled MRPs by M labeled MRPs should show comparable kinetics for MRPs of the same module. Hence, clustering of the normalized H and M labeled MRP abundances over time and fractions could provide information about the assembly pathway. However, several factors complicated data analysis: (1) MS data are not only noisy but the absence of a signal does not necessarily imply absence of the protein or peptide in an analyte; (2) spatial constraints must guide clustering analysis because not all MRPs can interact and form modules; and (3) not all MRPs were identified in the MS data, resulting in potential clustering gaps. Therefore, we first aimed to simplify (that is, reduce data dimensionality) the normalized H and M labeled MRP abundances across fractions over time (Fig. 1c and Supplementary Fig. 2) by computing fluxes of MRPs across sucrose gradient fractions as described below. Estimated fluxes for all MRPs together with their steady-state abundances (Supplementary Table 3) were then compared within all possible MRP groups considering spatial constraints (Supplementary Data 8 and 9). A contact matrix for mtSSU and mtLSU MRPs was computed, indicating the pairwise surface area. The latter was extracted on the basis of the structure PDB 6ZM6 using the software PDBePISA (Supplementary Fig. 4). Visual comparison of fluxes and abundances within each possible spatial group allowed deriving a first mtSSU and mtLSU assembly map. The latter was further refined and validated with targeted immunoprecipitation and knockout cell lines (Fig. 1b). We evaluated each cluster by computing cluster heterogeneity and comparing to all alternative clusters that could be derived solely using contact matrix constraints (Fig. 2, Extended Data Fig. 5 and Supplementary Tables 4 and 5). Cluster heterogeneity, $H$, was computed as the squared differences of fluxes and abundances between all cluster members over all sucrose gradient fractions proceeding and including the earliest fraction in which the cluster was assigned:

$$H_{\text{target},C,N_0}$$
$$= \frac{1}{N-N_0+1} \sum_{n=N_0}^{N} \left\{ \frac{1}{2} \left( mean_{i,j\in C} \left[ (f_{i,n} - f_{j,n})^2 \right] + mean_{i,j\in C} \left[ (q_{i,n} - q_{j,n})^2 \right] \right) \right\},$$

where $f$ and $q$ indicate the flux and abundance, respectively, of MRPs $i$ and $j$ in fraction $n$ and $N_0$ indicates the earliest sucrose gradient fraction in which the target MRP was assigned to its cluster $C$; $N$ was set to 7 for mtSSU and 8 for mtLSU. For each target MRP, we compared the heterogeneity of the selected cluster with the heterogeneity of a cluster consisting of all MRPs in contact with the target MRP (pink dots in Fig. 2 and Extended Data Fig. 5 and Supplementary Data 8 and 9). Furthermore, we determined the heterogeneity of all alternative clusters of the target MRP that consisted of the same number of cluster members as the selected cluster (violin plots and bee swarm plots in Fig. 2 and Extended Data Fig. 5). If the heterogeneity of the selected

cluster was the smallest among all comparisons, we considered this cluster as rather confident. However, wherever possible, best clusters were further challenged and validated with targeted immunoprecipitation and knockout cell line experiments. If the heterogeneity of the selected cluster was not the smallest compared to alternative clusters, we experimentally explored alternative clusters to either confirm the selected cluster or refine the clustering (Supplementary Tables 4 and 5, Fig. 2 and Extended Data Fig. 5). The final clusters, resulting in the constructed assembly pathways of the mtSSU and mtLSU, are illustrated as dendrograms in Supplementary Figs. 5 and 12, where the distance between an MRP node and a module node indicates the distance between the flux and abundance of the MRP compared to the mean of the fluxes and abundances of all MRPs assigned to the module. The distance, $d$, between an MRP and its module was defined as

$$d_{\text{mod},i} = \frac{1}{2} \left( (f_{\text{mod}} - f_i)^2 + (q_{\text{mod}} - q_i)^2 \right),$$

where $f_{\text{mod}}$ and $q_{\text{mod}}$ are the average flux and abundance, respectively, of the module in the assigned sucrose gradient fraction and $f_i$ and $q_i$ are the flux and abundance, respectively, of the MRP in the same sucrose gradient fraction. Hence, the larger the distance, $d$, the more divergent was the MRP from its assigned module. Accordingly, the distance between two module nodes indicates the distance between the average flux and abundance of the smaller module compared to the average of the fluxes and abundances of the larger module.

**Flux estimation of MRPs through sucrose gradient fractions.** We constructed a protein flux model, describing the transfer of H labeled MRPs through the sucrose gradient fractions. The aim of the protein flux model was to reduce the dimensionality of the full dataset to derive the mtSSU and mtLSU assembly pathways. The resulting rates from the flux model represent a simplified summary of various kinetic rates (for example, binding and unbinding rates) and, therefore, help only to characterize the structure of the assembly network; they do not reflect the physiological rates of the mitochondrial ribosome assembly pathway.

An H labeled protein in fraction $i$ can be degraded with rate $k_i$, be incorporated into a higher-molecular-weight protein complex with rate $a_i$ and, hence, migrate to fraction $i+1$ or dissociate into from its current complex and, hence, migrate to fraction $i-1$. For simplicity, we reduced this model by neglecting dissociation rates. The inclusion of dissociation rates in the flux model would result in an underdetermined system, in which many parameter combinations would result in the same model output. Therefore, including dissociation rates into the flux model would only increase model complexity without information gain but at the risk of less robust parameter inference. Including dissociation rates is also not necessary at this step because the flux model is only used as a data reduction approach. Therefore, for each MRP, we obtained the following system of ODEs:

$$dH_i(t)/dt = -(k_i + a_i) \times H_i(t) + a_{i-1}H_{i-1}(t),$$

where $H_i(t)$ is the normalized H labeled MRP abundance in sucrose gradient fraction $i$ at time $t$. For mtSSU related MRPs, $i$ ranges from 1 to 7 while, for mtLSU related MRPs, $i$ ranges from 1 to 8. The normalized M labeled MRP abundance in sucrose gradient fraction $i$ at time $t$ is then computed as $M_i(t) = H_i(t=0) - H_i(t)$ (Supplementary Fig. 3). The resulting flux model has a set of parameters $\theta = (k_i, a_i$ and $H_i(t=0))$ with $1 < i < 7$ for MRPs of the mtSSU (21 parameters) and with $1 < i < 8$ for MRPs of the mtLSU (24 parameters). The flux model was numerically solved using the 'lsoda' solver from the package 'deSolve' in R[62–64]. Initial conditions of $H_i(t)$ were inferred as model parameters $H_i(t=0)$, while initial conditions of $M_i(t)$ were set to $M_i(t=0) = 0$.

Normalized MRP abundances from triple-SILAC MS data were used to infer all model parameters to investigate the observed kinetic behavior of MRPs across sucrose gradient fractions. During parameter inference, data points of a given fraction were only considered if, for at least three time points, finite ratios larger than zero could be computed between H and M isotopes. In this way, fluxes and abundances were estimated only for fractions in which MRPs were robustly detected.

Parameters were estimated applying a Bayesian approach as originally proposed in a previous study[65]. Briefly, the posterior distribution $p(\theta|D)$ of the parameter vector $\theta$ is defined as

$$p(\theta|D) = \frac{p(D|\theta) \cdot p(\theta)}{p(D)},$$

where $p(\theta)$ is the prior distribution of the parameters $\theta$ and $p(D|\theta)$ is the likelihood of the data $D$ given the parameters $\theta$. The aim is to find a set of parameters $\theta$ that maximize the likelihood $p(D|\theta)$ (Supplementary Fig. 9). Here, the log likelihood was defined as

$$\ln(p(D|\theta)) = \sum_i \sum_t \ln\left(L_{i,t}^1\right) + \ln\left(L_{i,t}^2\right),$$

with

$$L_{i,t}^1 = p(H_i(t)|\theta) \sim \mathcal{N}(\mu = H_i(t), \sigma = \text{s.d.})$$

and

$$L_{i,t}^2 = p(M_i(t)|\theta) \sim \mathcal{N}(\mu = M_i(t), \sigma = \text{s.d.}),$$

where $\mathcal{N}$ indicates the probability density of the normal distribution with mean $\mu$ and s.d. $\sigma$. The s.d. is inferred as an additional parameter. Parameter inference was performed using the BayesianTools R package[66]. A uniform prior $p(\theta) \sim \mathcal{U}([\min, \max])$ was used to infer the model parameters. Uniform prior ranges (min, max) are displayed in Supplementary Table 8. Differential evolution Markov chain Monte Carlo (DE-MCMC) with $Z$ past steps and Snooker update (zs) sampler implemented in $R$ was applied[28]. Parameters were inferred using three start values, a Snooker update probability of 0.001, a thinning parameter of 10 and a multiplicative error of 0.2. The scaling factor $\gamma$ was kept at 2.38, setting it to 1 with a probability of 0.1. The posterior distribution for each MRP was saved and diagnostic plots were obtained. Inference was run for $10^6$ iterations. Convergence was manually inspected for all MRPs.

For each MRP, posterior parameter distributions were obtained for the flux model. Sampling 1,000 particles from this distribution allowed us to compute the median and the 5th and 95th percentile confidence ranges of the fluxes, $f_i$, as $f_i = k_i + a_i$ (Supplementary Data 6 and 7 and Supplementary Table 3) for each considered sucrose gradient fraction $i$.

**mtSSU kinetic modeling.** The constructed mtSSU assembly pathway displayed in Fig. 3 provided a basis to construct a kinetic mtSSU assembly model. In the absence of molecular knowledge of the precise interactions during mtSSU assembly, we used simple mass action kinetics to derive a system of ODEs with parameters representing kinetic rates of the assembly pathway. Estimation of those kinetic rates based on the triple-SILAC MS data allowed gaining insight about into module stability and critical assembly steps (Extended Data Fig. 4). Below, we provide details of the mtSSU kinetic model, Bayesian parameter inference and parameter sensitivity analysis.

The mtSSU assembly model consists of single MRPs, some of which were also detected in early sucrose gradient fractions, and mtSSU modules, which consisted of two or more MRP members. First, we

constructed a dataset based on the normalized abundances of H and M labeled proteins ($H_i(t)$ and $M_i(t)$) across all fractions, describing the kinetic behavior of both single MRPs and mtSSU modules. The abundance of single MRPs was extracted directly from the normalized abundances of H and M labeled proteins ($H_i(t)$ and $M_i(t)$). The abundance of modules was computed as the mean over normalized abundances of all MRPs assigned as a member of the module (Supplementary Table 9) for each time point. The resulting kinetics of H and M labeled proteins are displayed in Supplementary Fig. 10.

With the reconstructed mtSSU assembly pathway at hand (Fig. 3), we next derived a system of ODEs describing five possible reactions involving mtSSU modules and four reactions involving single MRPs. A given MRP can (1) be imported into mitochondria (supply); (2) be degraded as such; (3) be incorporated into an mtSSU module (binding); or (4) dissociate from an mtSSU module to a single MRP protein (unbinding). Therefore, each MRP takes part in four reactions. The 12S rRNA is considered the same way as all remaining MRPs. A given mtSSU module can (1) be degraded as such (turnover); (2) be incorporated into a larger-molecular-weight mtSSU module (binding); (3) dissociate from a larger-molecular-weight mtSSU module (unbinding); (4) be generated through binding of MRPs to lower-molecular-weight mtSSU modules (binding); or (5) dissociate into lower-molecular-weight mtSSU modules and MRPs (unbinding), resulting in five reactions (Extended Data Fig. 4a). Each of these reactions occurs with kinetic rates (that is, supply rates, turnover rates, binding rates and unbinding rates). In absence of detailed knowledge about the assembly steps, we described the mtSSU kinetic model using mass action kinetics in generalized form. The reaction network leading to the assembly of the full mtSSU can be broken down into single reactions, each having one reactant E and one product P. The integration of several MRPs leads to the formation of the same mtSSU module (as depicted in Extended Data Fig. 4a). This system can be represented by a set of equations with different reactants E and the same product P (Supplementary Table 10).

Iterating over all products indexed 1 to $j$, each resulting as a product from all reactants $k$, results in the following set of ODEs:

$$\frac{d[P_j]}{dt} = \text{on}_{P_j} \prod_k [E_k] - \text{off}_{P_j}[P_j], \tag{1}$$

$$\frac{d[E_k]}{dt} = -\text{on}_{P_j} \prod_k [E_k] + \text{off}_{P_j}[P_j], \tag{2}$$

where $\text{on}_{P_j}$ is the binding rate of a given product (mtSSU module), $\text{off}_{P_j}$ is the unbinding rate of the mtSSU module and [.] indicates the normalized abundance of the respective component.

This procedure results in a partial derivative $\frac{\partial[C]}{\partial t}$ for all components C (MRPs and mtSSU modules) of the model, comprising all reactants and products:

$$\frac{\partial[C]}{\partial t} = \sum_{j \in j_C} \frac{d[P_j]}{dt} + \sum_{k \in k_C} \frac{d[E_k]}{dt}, \tag{3}$$

with $j \in j_C$ depicting all reactions where C is a product and $k \in k_C$ depicting all reactions where C is a reactant. In addition, all components are turned over (degraded) with the rate $(k_C)$; hence, the ODE for a single model component C (MRPs and mtSSU modules) can be expressed as the sum of Equation (3) and the turnover:

$$\frac{d[C]}{dt} = \frac{\partial[C]}{\partial t} - k_C[C]. \tag{4}$$

This model describes the full mtSSU kinetic model of total MRPs and mtSSU module abundances. Because the total abundance of all

model components is constant over time because of steady-state conditions, we can rearrange Equations (1)–(4) to derive expressions for all MRP supply rates ($\sup_{C_{mrp}}$):

$$\sup_{C_{mrp}} = -\frac{d[C_{mrp}]}{dt},$$

with $C_{mrp}$ being the model components describing MRPs. We furthermore rearrange Equations (1)–(4) to derive expressions for all mtSSU module turnover rates ($k_{C_{module}}$):

$$k_{C_{module}} = \frac{\partial[C_{module}]}{\partial t} \cdot \frac{1}{[C_{module}]},$$

where $C_{module}$ describes all mtSSU modules. $\frac{\partial[C_{module}]}{\partial t}$ is derived as described in Equation (3). Those parameters were considered as dependent parameters to fulfill the steady-state condition. All remaining parameters were inferred from the data, as described below. Therefore, the dimension of the unknown parameter space describing kinetic rates was reduced from 111 to 64 parameters.

To model the kinetics of H and M labeled MRPs and mtSSU modules, we reformulated the system of ODEs defined in Equations (1)–(4). Firstly, instead of modeling total protein abundance, we modeled the abundance of H labeled proteins. Secondly, the supply rates of the MRPs were set to 0 (that is, no newly synthetized H labeled MRPs were imported into mitochondria upon growth media exchange from H to M). Thirdly, the initial conditions of H labeled MRPs and mtSSU modules ($H_i(t = 0)$) corresponded to their total steady-state abundance but were inferred as additional parameters. Solving this system of ODEs given a set of parameters and initial conditions allowed obtaining abundances for H labeled MRPs and mtSSU modules over time ($H_i(t)$). The abundance of M labeled MRPs and mtSSU modules ($M_i(t)$) was computed as $M_i(t) = H_i(t = 0) - H_i(t)$.

The resulting mtSSU model has a set of unknown parameters $\theta = (k_i, \text{on}_j, \text{off}_j \text{ and } P_l(t = 0))$ with $1 < i < 30$ describing the turnover rates of the single MRPs, $1 < j < 17$ describing the binding and unbinding rates for the mtSSU modules and $1 < l < 47$ describing the initial conditions of single MRP abundances and mtSSU module abundances (111 parameters).

The mtSSU kinetic ODE model was implemented in matrix form and numerically solved using the 'lsoda' solver from the package deSolve (version 1.35) in R[62–64]. Packages stringr (version 1.5.0) and dplyr (version 1.1.1) were used for data processing. The initial conditions of $P_i(t)$ were inferred as model parameters, while the initial conditions of $M_i(t)$ were set to $M_i(t = 0) = 0$.

Normalized MRP abundances for H and M labeled proteins ($H_i(t)$ and $M_i(t)$), as well as their log ratios $R_i(t) = \log(M_i(t)/H_i(t) + 1)$ from triple-SILAC MS data, were used to infer all model parameters. As for the kinetic mtSSU model, parameters were estimated applying Bayesian inference using the BayesianTools (version 0.1.8) R package[66]. A truncated normal prior distribution $p(\theta) \sim N([\mu, \sigma, \min = 0, \max])$ was used to infer the model parameters. Truncated normal prior parameters ($\mu$, $\sigma$ and max) are displayed in Supplementary Table 11. As for the kinetic mtSSU model, DE-MCMC with $Z$ past steps and Snooker update ($zs$) sampler implemented in $R$ was applied[28]. Parameters were inferred using three start values, a Snooker update probability of $1 \times 10^{-3}$, a thinning parameter of 9 and a multiplicative error of 0.5. The scaling factor $\gamma$ was kept at 2.38, setting it to 1 with a probability of $1 \times 10^{-3}$. The posterior distribution for each MRP was saved and diagnostic plots were obtained. Inference was run for $3 \times 10^{4}$ iterations. Convergence was manually inspected. The log likelihood of the mtSSU kinetic model was defined as follows:

$$\ln(p(D|\theta)) = \sum_i \sum_t \ln\left(L^1_{i,t}\right) + \ln\left(L^2_{i,t}\right) + + \ln\left(L^3_{i,t}\right),$$

with

$$L^1_{i,t} = p(H_i(t)|\theta) \sim \mathcal{N}(\mu = H_i(t), \sigma = \text{s.d.}),$$

$$L^2_{i,t} = p(M_i(t)|\theta) \sim \mathcal{N}(\mu = M_i(t), \sigma = \text{s.d.})$$

and

$$L^3_{i,t} = p(R_i(t)|\theta) \sim \mathcal{N}(\mu = R_i(t), \sigma = \text{s.d.}),$$

where $\mathcal{N}$ indicates the probability density of the normal distribution with mean $\mu$ and s.d. $\sigma$. The s.d. was inferred as additional parameter (resulting in 112 total parameters). The dependent parameters were computed on the basis of the inferred parameters for each sampling iteration during inference. If any of those dependent parameters was smaller than 0, the likelihood was set to a strongly negative value, resulting in rejection of this parameter combination.

Priors of initial conditions $P_i(t = 0)$ of those single MRPs and mtSSU modules that were detected in a given fraction were modified according to their measured normalized abundance. The respective prior mean and s.d. values were set to the mean and s.d. of the measured normalized abundance. Lower and upper prior boundaries were set to 75% and 125% of the means, respectively.

The obtained posterior parameter distribution provides insights into the kinetics of the assembly steps. Marginal posterior parameter distributions of the kinetic rates and the initial conditions of the MRP and modules (Supplementary Fig. 11 and Supplementary Table 6) were obtained by sampling 1,000 particles from the posterior distribution.

**Local sensitivity analysis.** To investigate which mtSSU assembly steps were most influential on mtSSU steady-state abundance, local sensitivity analysis was performed. A total of 1,000 particles from the posterior parameter distribution of the mtSSU kinetic model were sampled and used to simulate the full mtSSU steady-state model until $t = 48$ h. The simulated mtSSU steady-state abundance was considered as the baseline. Model simulation was then repeated for all 1,000 particles, while multiplying a given parameter by a factor $x$, where $x$ ranges from $10^{-4}$ to $10^{4}$, and resulting mtSSU abundances after 48 h were recorded and compared to the baseline. In this way, we computed the fold change of mtSSU steady-state abundance upon a fold change of a given parameter across all 1,000 particles. Median fold changes were computed and are displayed in Extended Data Fig. 4. Hierarchical clustering of computed fold changes across MRPs was performed using the function 'heatmap' in R (with ward.d method in clustering function hclust using Euclidean distance) for visualization.

**Formation of assembly modules is independent of the presence of rRNA (inference of decay rates upon mt-rRNA depletion upon ethidium bromide treatment).** The MRP decay rate was determined upon repression of mt-rRNA synthesis by ethidium bromide. An exponential decay model was implemented and used to infer decay rates ($k$) of MRPs and 12S RNA:

$$p(t) = 100e^{-kt},$$

where $p$ is the percentage of protein detected at time $t$ relative to $t = 0$. The decay rate was inferred using Bayesian inference (as described above) with uniform prior distribution (min = 0, max = 2). DE-MCMC with $Z$ past steps and Snooker update ($zs$) sampler implemented in $R$ was applied[28]. Parameters were inferred using three start values, a Snooker update probability of $1 \times 10^{-3}$, a thinning parameter of 9 and a multiplicative error of 0.5. The scaling factor $\gamma$ was kept at 2.38, setting it to 1 with a probability of $1 \times 10^{-3}$. The posterior distribution for each

MRP was saved and diagnostic plots were obtained. Inference was run for $3 \times 10^4$ iterations. The protein's half-life was calculated as follows:

$$t_{1/2} = \frac{\ln(2)}{k}.$$

**MS data analysis of assembly factors.** Protein intensities, derived from MS data, were extracted for all detected assembly factors across sucrose gradient fractions and over time. H and M labeled proteins were considered. The L spike-in as a standard was not considered here. In the steady state, assembly factors should have constant abundance over time. Hence, the sum of H and M labeled protein at each time point should be constant within measurement noise. Therefore, we determined the steady-state abundance of a protein by computing the mean over summed H and M labeled proteins over time for each fraction. Steady-state abundances were scaled per protein by division with the maximal steady-state abundance of the protein.

### Genetic engineering in cell culture
Generation of HEK293 knockout cell lines was performed as described previously[67] using Alt-R clustered regularly interspaced short palindromic repeats (CRISPR)–Cas9 technology (Integrated DNA Technologies). In brief, cells were cotransfected with a CRISPR RNA (crRNA)–trans-activating crRNA duplex and Cas9 nuclease and single cells were separated by fluorescence-activated cell sorting (FACS). The sequences of crRNAs targeting genes encoding selected MRPs are listed in Supplementary Table 12. Obtained single cell-derived clones were screened by immunoblotting and verified by Sanger sequencing.

Stable tetracycline-inducible HEK293 cell lines expressing C-terminal FLAG-tagged MRPs were generated according to the established protocol[68]. pOG44 and pcDNA5/FRT/TO plasmids containing the respective FLAG construct were delivered by lipofection using Lipofectamine 3000 according to the manufacturer's instructions. Clones were selected with hygromycin B (100 μg ml$^{-1}$) and blasticidin S (5 μg ml$^{-1}$) (Gibco) and the expression of the FLAG-tagged protein was confirmed by western blotting.

### Depletion of mtDNA
Ethidium bromide treatment was used to inhibit mtDNA replication and transcription followed by depletion of mtDNA-encoded RNAs including 12S rRNA (mtSSU), 16S rRNA and tRNA$^{Val}$ (mtLSU)[26,27]. HEK293 cells were grown in the presence of 0.25 μg ml$^{-1}$ ethidium bromide (Roth) for indicated time points to estimate the MRPs and mtDNA-encoded RNAs half-lives (for example, Fig. 4b) or for 48 h for immunoprecipitation or sucrose gradient analysis.

### Immunoprecipitation of mitoribosomal complexes
Mitochondria (6 mg for sucrose gradient sedimentation and 1 mg for western blot analysis) isolated from stable inducible cell lines bearing a FLAG-tagged MRP construct were lysed in lysis buffer (100 mM NH$_4$Cl, 15 mM MgCl$_2$, 20 mM Tris-HCl pH 7.5, 1% (w/v) digitonin, 10% (w/v) glycerol, 1 mM PMSF, 0.08 U per μl RiboLock RNase Inhibitor (Thermo Fisher) and cOmplete protease inhibitor cocktail (Roche)). Cleared supernatant (16,000g at 4 °C for 10 min) was subjected to coimmunoprecipitation using anti-FLAG M2 Affinity Gel (Sigma) for 1 h. The beads were washed thoroughly seven times with wash buffer (100 mM NH$_4$Cl, 15 mM MgCl$_2$, 20 mM Tris-HCl pH 7.5, 0.2% (w/v) digitonin, 10% (w/v) glycerol, 1 mM PMSF, 0.08 U per μl RiboLock RNase Inhibitor (Thermo Fisher) and cOmplete protease inhibitor cocktail (Roche)) followed by three washes with buffer without glycerol and PMSF for the subsequent sucrose gradient ultracentrifugation. For the western blot analysis, the beads were washed ten times with wash buffer with all additives. Elution of copurified mitoribosomal complexes was achieved by addition of FLAG peptide (Sigma) in wash buffer without glycerol and PMSF at 1,000 r.p.m. shaking at 4 °C for 30 min. The eluates were immediately used for sucrose gradient sedimentation or mixed with SDS loading buffer (10% (v/v) glycerol, 2% (w/v) SDS, 0.01% (w/v) bromophenol blue, 63 mM Tris-HCl pH 6.8 and 5 mM DTT) and resolved onto 10–18% SDS polyacrylamide gel.

### [$^{35}$S]methionine *de novo* incorporation
Monitoring of the mitochondrial translation was performed according to the established protocol[69]. After inhibiting the cytosolic translation with 100 μg ml$^{-1}$ emetine (Merck), cells were treated with 0.2 mCi ml$^{-1}$ [$^{35}$S]methionine for 1 h and washed three times with PBS. Cell pellets were lysed in lysis buffer (50 mM Tris-HCl pH 7.4, 130 mM NaCl, 2 mM MgCl$_2$, 1% NP-40, 1 mM PMSF and cOmplete protease inhibitor cocktail (Roche)), and cleared lysates were subjected to SDS–PAGE followed by western blotting. Radiolabeled mitochondrial translation products were visualized by the Typhoon imaging system (GE Healthcare).

### SDS–PAGE and western blotting
Cell lysates or proteins recovered from sucrose gradient fractions were mixed with SDS loading buffer (10% (v/v) glycerol, 2% (w/v) SDS, 0.01% (w/v) bromophenol blue, 63 mM Tris-HCl pH 6.8 and 5 mM DTT). Samples were resolved using 10–18% SDS–PAGE. Proteins were transferred onto a nitrocellulose membrane (Cytiva) and analyzed by western blotting. After blocking against nonspecific binding in 5% (w/v) milk, the membranes were incubated overnight with specific primary antibodies at 4 °C. After treatment with horseradish peroxidase-conjugated secondary antibodies, the signal was detected using ECL western blotting substrate (ThermoFisher Scientific). The primary and secondary antibodies used in this study are listed in Supplementary Table 12.

### RNA extraction and northern blotting
RNA was isolated from cultured cells or sucrose gradient fractions following the standard TRIzol (Life Technologies) extraction protocol using GlycoBlue coprecipitant (Invitrogen). Extracted RNA was separated by denaturing formaldehyde–formamide gel containing 1.2% (w/v) ultrapure agarose (Invitrogen) and transferred to Amersham Hybond-N membrane (GE Healthcare). Mitochondrial rRNAs were detected by $^{32}$P-radiolabeled probes targeting *MTRNR1* (12S), *MTRNR2* (16S) or 18S rRNA (Supplementary Table 12) and visualized by the Typhoon imaging system (GE Healthcare).

### Statistics and reproducibility
All details on statistics and reproducibility of the MS data are specified in Methods, 'MS data analysis'.

Experiments that included immunoprecipitation of mitoribosomal complexes from cell lines bearing FLAG-tagged MRP constructs with subsequent sucrose gradient sedimentation analysis were conducted to verify the composition of the mitoribosome assembly clusters gained from MS data. These immunoprecipitation experiments were performed once or twice when there were no alternative clusters for a given bait protein (for example, mS22$^{FLAG}$ and mS27$^{FLAG}$) and at least in triplicate with similar results when MS-derived clusters required further elaboration (for example, bS1m$^{FLAG}$ and mL62$^{FLAG}$) (Supplementary Tables 4 and 5).

Sucrose gradient sedimentation analysis of mitoribosomal complexes isolated from knockout cell lines was performed at least three times for each cell line, consistently yielding the same results.

Experiments determining steady-state levels of MRPs and formation of mitoribosome assembly intermediates upon mtDNA-encoded rRNA depletion (ethidium bromide treatment) were successfully replicated three times.

### Reporting summary
Further information on research design is available in the Nature Portfolio Reporting Summary linked to this article.

## Data availability

Materials are available upon reasonable request. The original data generated in this study are provided in the Supplementary Information and on Figshare (https://doi.org/10.6084/m9.figshare.25382326). The MS original data, protein sequence databases (downloaded from UniProt Knowledgebase), MaxQuant analysis files and database search output files were deposited to the MASSIVE repository and are available using the accession codes MSV000091653 and MSV000091652. Source data are provided with this paper.

## Code availability

R Scripts and Python scripts for data processing and visualization are provided on Figshare (https://doi.org/10.6084/m9.figshare.25382326).

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

## Acknowledgements

We thank M. Bohnsack, P. Rehling, K. Bohnsack and I. Wohlgemuth for critical reading of the manuscript and M. Raabe for technical support. We thank P. Rehling (University Medical Center Göttingen) for providing antibodies. This work was funded and supported by the German Research Foundation (DFG, Deutsche Forschungsgemeinschaft) under Germany's Excellence Strategy (EXC 2067/1-390729940 to R.R.-D. and H.U.), the DFG Emmy-Noether grant (RI 2715/1-1 to R.R.-D.), SFB860 (to R.R.-D. and H.U.), SFB1565 (project number 469281184 to R.R.-D. and H.U.), ERC-StG 945528 IMAP (to J.L., Y.H., M.P. and H.P.R.), the Max Planck Society (to J.L. and H.U.), the International Max Planck Research School (IMPRS) for Genome Science (to Y.H. and M.P.) and the King's College London as part of the Neuro-Immune Interactions in Health and Disease Wellcome Trust PhD Programme (to H.P.R.).

## Author contributions

Conceptualization, R.R.-D., E.L., J.L. and H.U.; investigation—cell culture and biochemical approaches, E.L., E.H., V.C., F.N., E.S., M.H. and M.M.-Q.M.; MS measurements, A.L. and L.W.; mathematic modeling, J.L. and M.P.; unfractionated MS data analysis, M.P.; mtSSU model implementation, H.P.R.; MS data processing and analysis, J.L., Y.H. and M.P.; writing—original draft, E.L., J.L. and R.R.-D.; writing—review and editing, E.L., J.L., H.U., R.R.-.D, M.P. and H.P.R.; visualization, E.L., V.C. and J.L.; supervision, R.R.-D., J.L. and H.U.

## Competing interests

The authors declare no competing interests.

## Additional information

**Extended data** is available for this paper at https://doi.org/10.1038/s41594-024-01356-w.

**Correspondence and requests for materials** should be addressed to Henning Urlaub, Juliane Liepe or Ricarda Richter-Dennerlein.

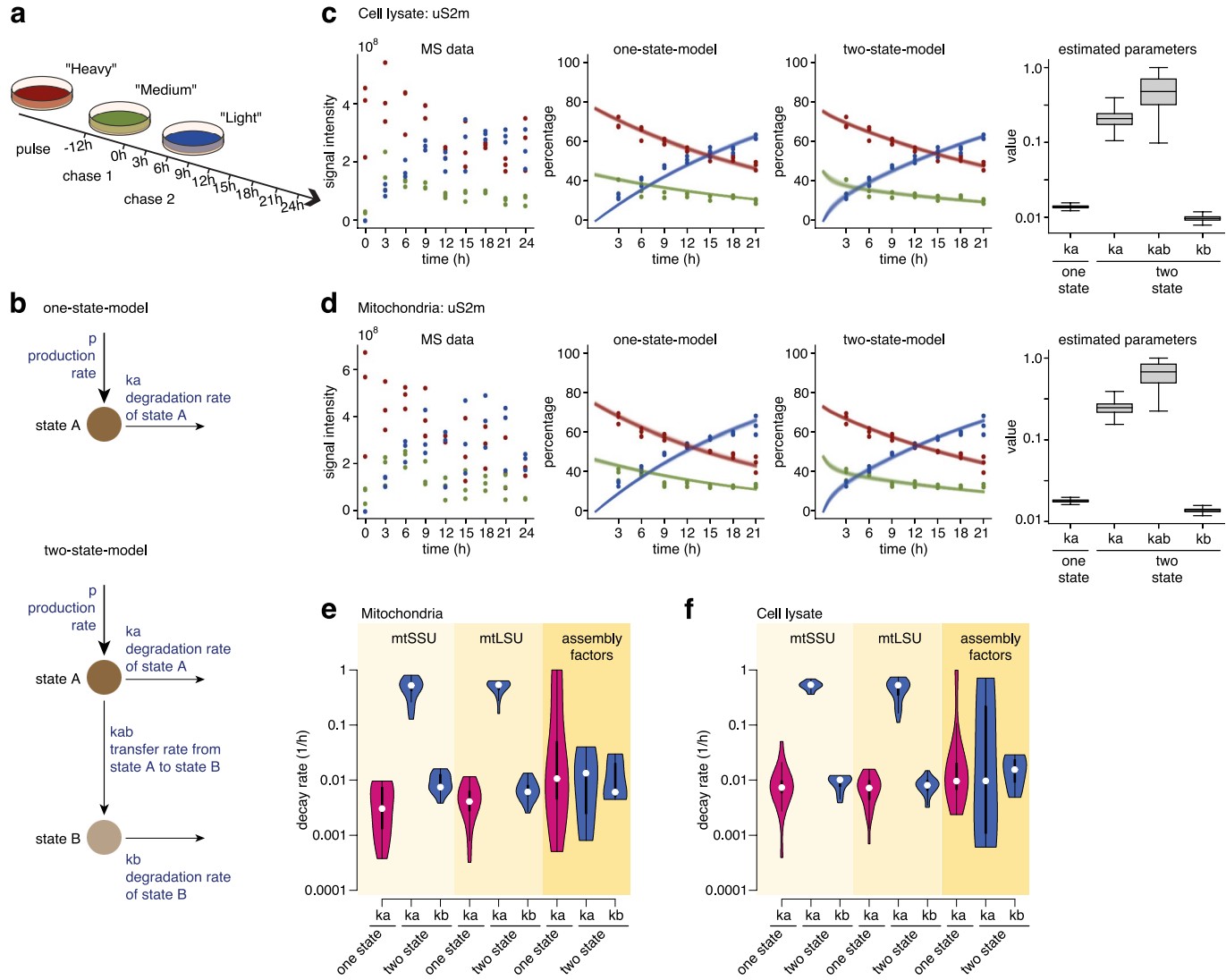

**Extended Data Fig. 1 | Global intracellular turnover of MRPs and mitoribosome assembly factors. a**, Triple SILAC approach. HEK293 cells were pulse labelled with 'heavy' amino acids (red) and then first chased with 'medium' amino acids (green) for 12 h, followed by a second chase with 'light' labelled amino acids (blue) for indicated time intervals (0h-24h). Samples were then analyzed by MS. **b**, Schematics of the employed models. In the one-state-model proteins are produced with rate $p$ and degraded with rate $k_a$. In the two-state-model proteins are also transferred from state $A$ to state $B$ with rate $k_{ab}$ and proteins in state $B$ are degraded with rate $k_b$. **c**, **d**, Raw mass spectrometry data, model fits of normalized data and posterior distribution of model parameters of protein uS2m (as example) in whole cell lysate (**c**) and isolated mitochondria

(**d**). Time points 0 and 24 are omitted from modelling for their inconsistency with other time points (**Methods: 'Protein turnover estimation – no sucrose gradient'**). State A (in two-state-model) represents transient state with faster turn-over while state B is more stable (based on the values of $k_a$, $k_{ab}$ and $k_b$). This allows the model to have different short-term and long-term dynamics. See medium and light isotopes of two-state-model in early vs late time-points. Solid lines indicate median and shaded areas indicate 5%-ile and 95%-ile of model fits. Boxplots indicate median, 1st quartile, 3rd quartile, as well as minimum and maximum after outlier removal. **e**, **f**, Global distribution (across protein groups) of parameters derived from isolated mitochondria (**e**) and whole cell lysate (**f**). In (**c**–**f**) n = 3 biological replicates were used for parameter estimation.

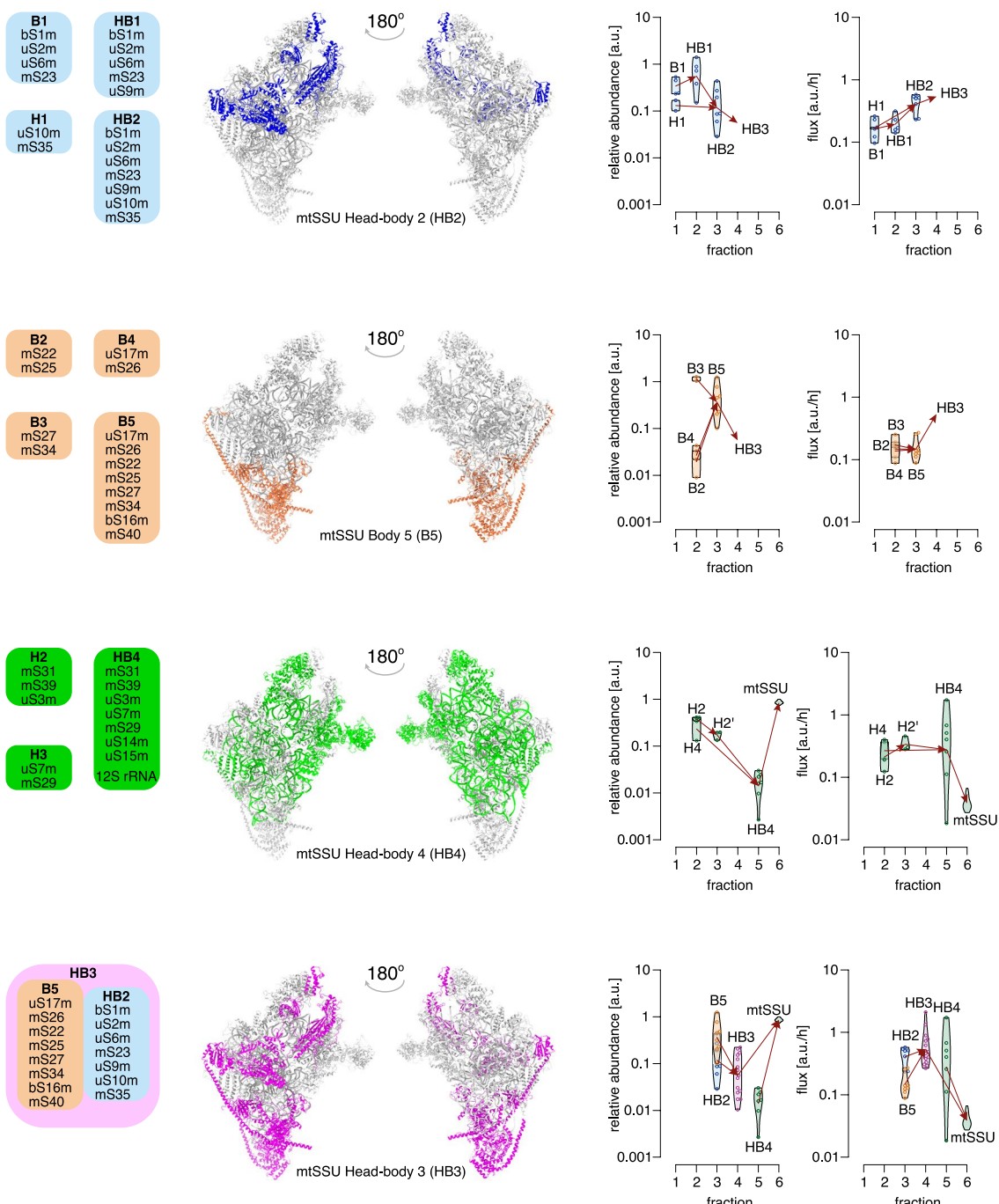

**Extended Data Fig. 2 | Clustering of mtSSU assembly modules.** Left-hand panel: schematic representation of the mtSSU assembly modules and their position in the mature mtSSU (PDB: 6zm6). Right-hand panel: clustering reveals 12 distinct mtSSU assembly modules. Shown are violin plots illustrating the distribution of estimated fluxes and abundances over mtSSU MRPs that are grouped into the same assembly module depending on the sucrose gradient fraction they were detected in. Arrows indicate the further assembly direction of a submodule. Color labels correspond to left panel. H – mtSSU head; B – mtSSU body; HB – mtSSU head-body assembly module.

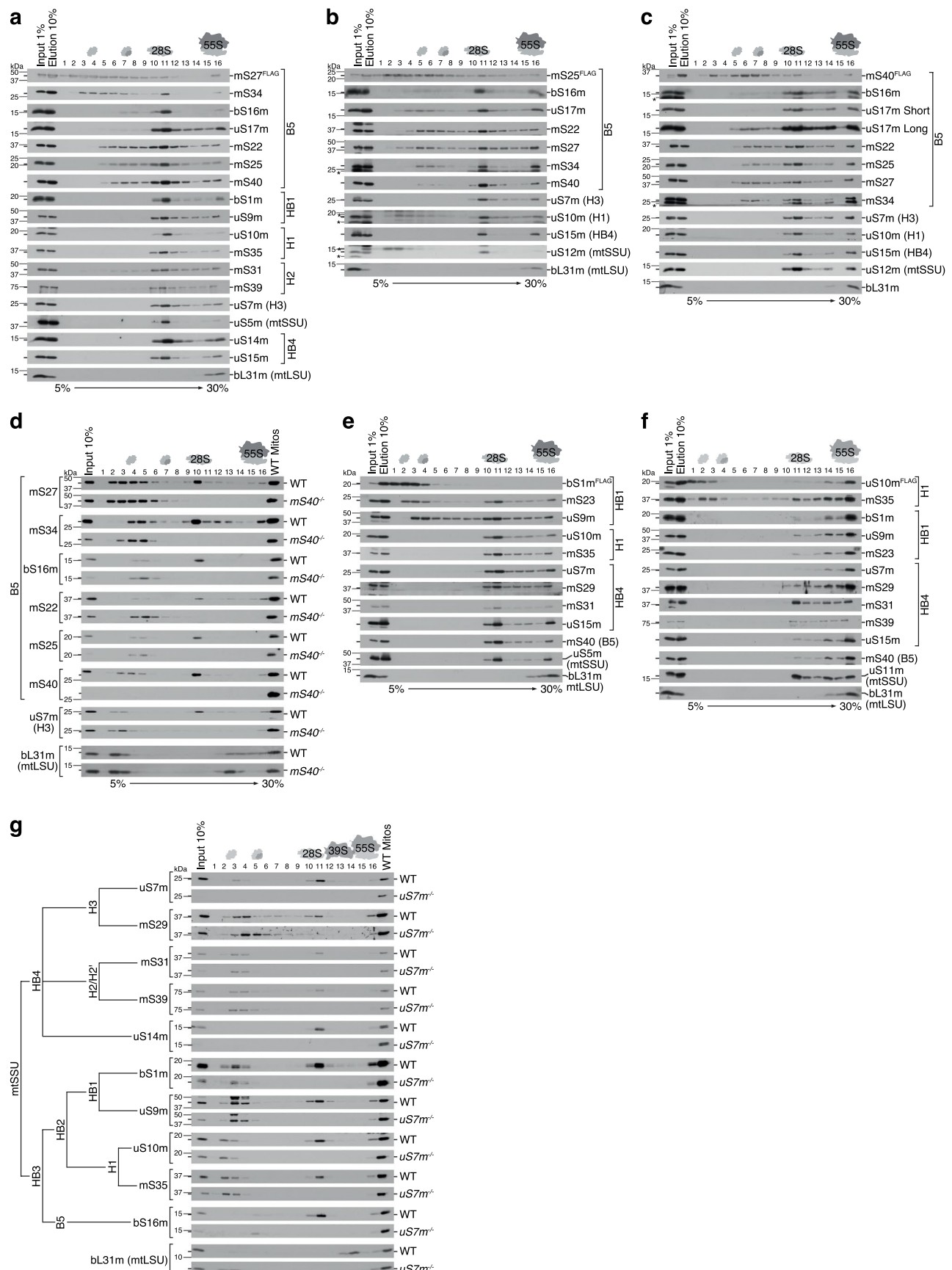

**Extended Data Fig. 3 | Biochemical validation of mtSSU assembly modules. a-c,** and **e-f,** The composition of the B5 (**a-c**), HB1 (**e**), and H1 (**f**) mtSSU assembly clusters. Mitoribosomal complexes were immuno-isolated via FLAG-tagged constituents mS27 (**a**), mS25 (**b**), mS40 (**c**), bS1m (**e**), and uS10m (**f**) and separated by sucrose gradient centrifugation. **d, g,** Sucrose gradient sedimentation analysis of mitoribosomal complexes in mS40 (**d**) or uS7m (**g**) deficient cells. H – mtSSU head; B – mtSSU body; HB – mtSSU head-body assembly module.

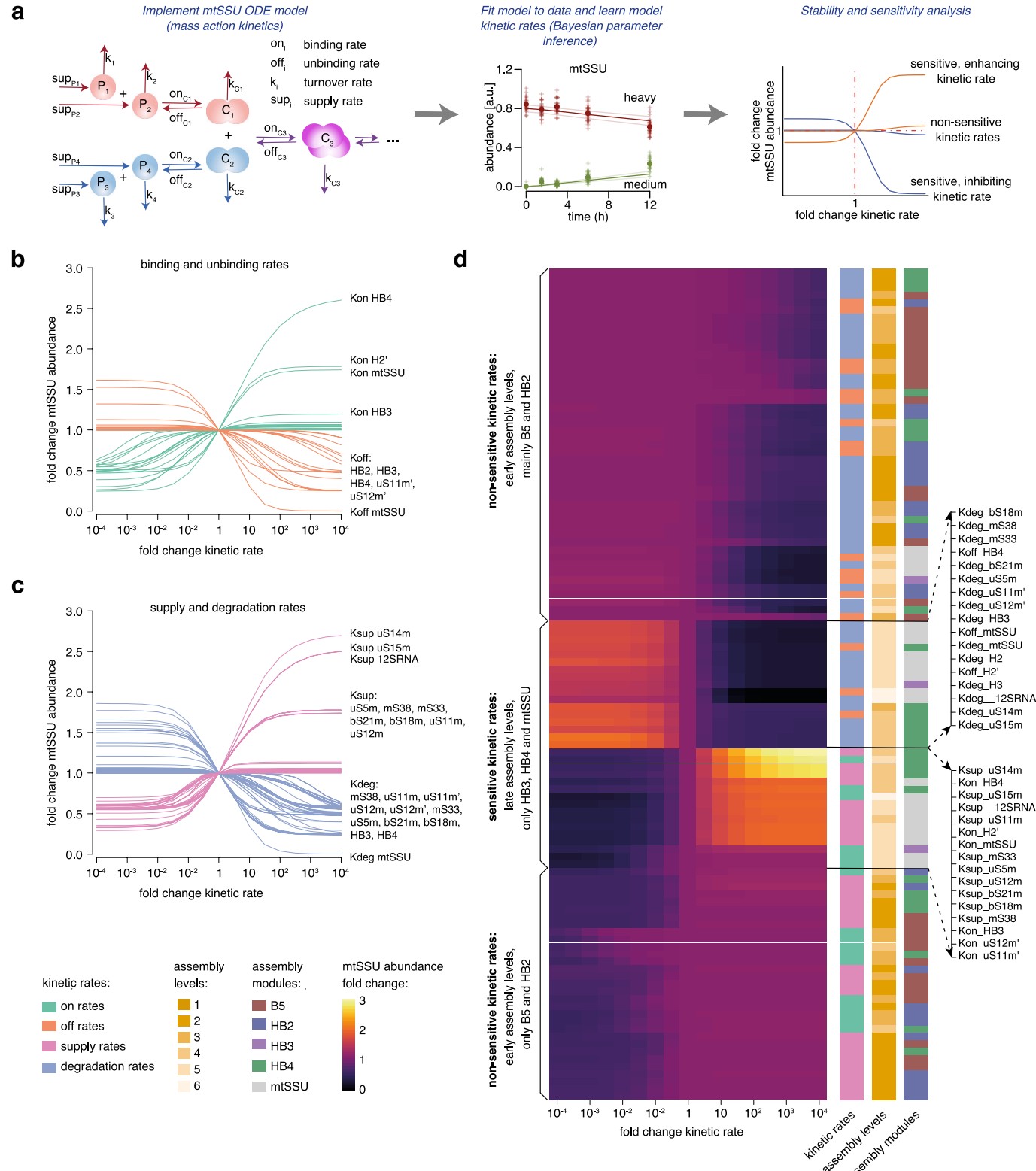

**Extended Data Fig. 4 | Kinetics of mtSSU assembly pathway. a**, Overview of mtSSU kinetic modelling and sensitivity analysis. The kinetic model consists of four reactions for each MRP: supply (sup) and turnover (k) rates indicating transport of MRPs into mitochondria and removal of MRPs from mitochondria (MRP recycling), binding (on) and unbinding (off) rates indicating incorporation of MRPs into complexes (association) and their dissociation. Bayesian inference was employed to estimate the most likely values for each kinetic rate in the full model (**Methods: '*mtSSU kinetic modelling*').** (Solid lines indicate median and shaded areas indicate 5%-ile and 95%-ile of model fits.) Finally, local sensitivity analysis was performed to determine the rates, which have the strongest impact

on mtSSU steady state abundance by changing a given rate and simulating the resulting mtSSU steady state abundance. Comparison of resulting mtSSU abundance fold change, allows to detect non-sensitive and sensitive kinetic rates. Latter can be either enhancing or inhibiting rates. **b**, Fold changes of mtSSU steady state abundance upon increase or decrease of binding and unbinding rates. **c**, Fold changes of mtSSU steady state abundance upon increase or decrease of supply and degradation rates. **d**, Hierarchical clustering reveals sensitive and non-sensitive kinetic rates. Shown are the fold changes of mtSSU steady state abundance, indicated as a range from black (minimal value) to light yellow (maximal value), for all kinetic rates.

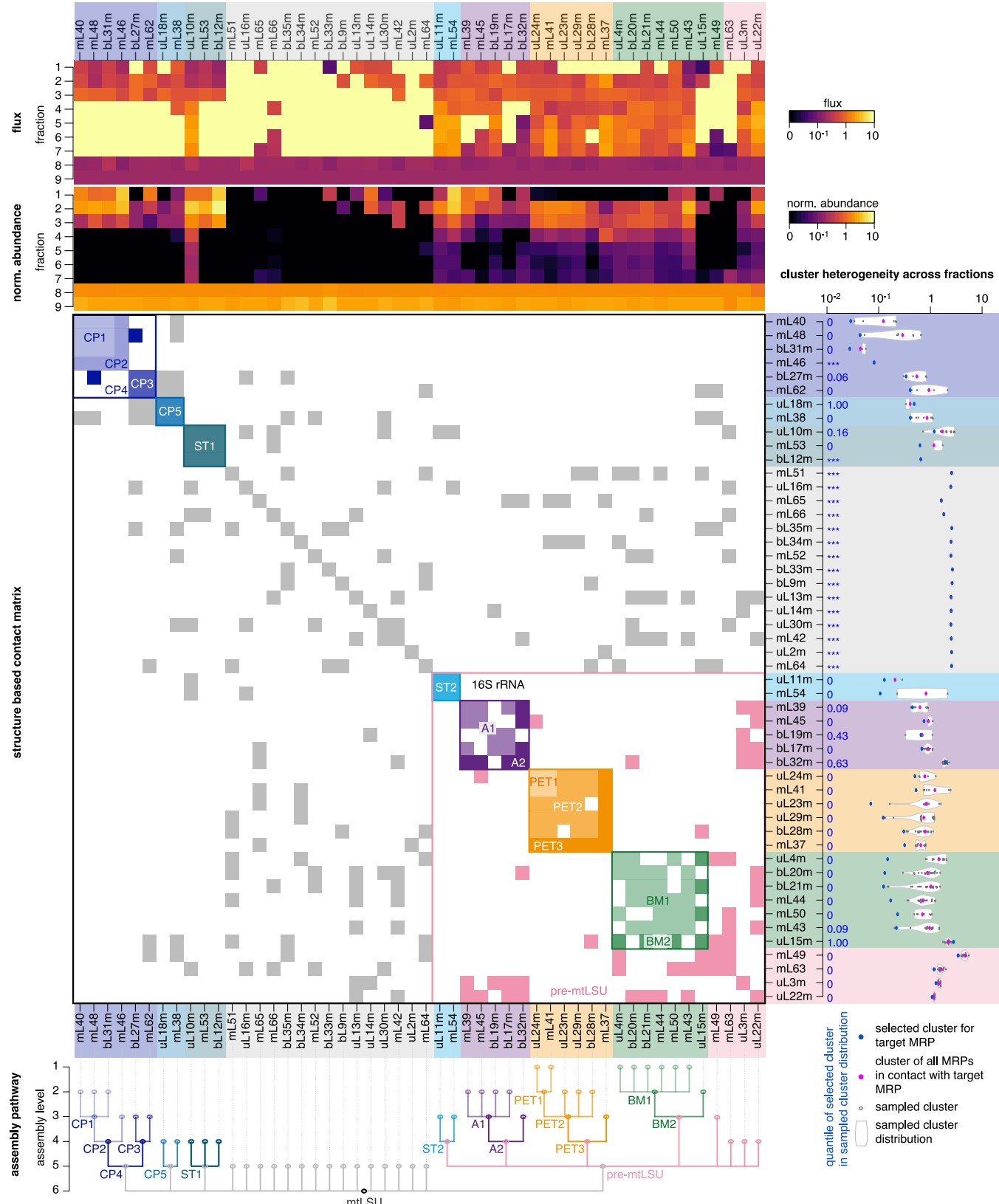

**Extended Data Fig. 5 | Cluster analysis of mtLSU assembly pathway.** Fluxes and normalized steady state abundances of all MRPs are shown as heatmap (top). mtLSU MRPs are aligned according to their clusters. The contact matrix in the center shows all pairwise MRP contacts derived from the known structure of the assembled mtLSU, colored according to their assigned assembly module. Cluster heterogeneity is indicated (right panel) for each target MRP as blue dots and compared to the heterogeneity of all possible clusters based only on contact matrix constraints (grey dots and violin plots). Blue indicated numbers define the quantile of the selected target cluster (blue dot) within the alternative cluster distribution, where 0 indicates that the selected target cluster has the lowest heterogeneity and *** indicates absence of alternative clusters. The resulting assembly pathway is shown as dendrogram with indicated assembly levels for all MRPs and mtLSU modules.

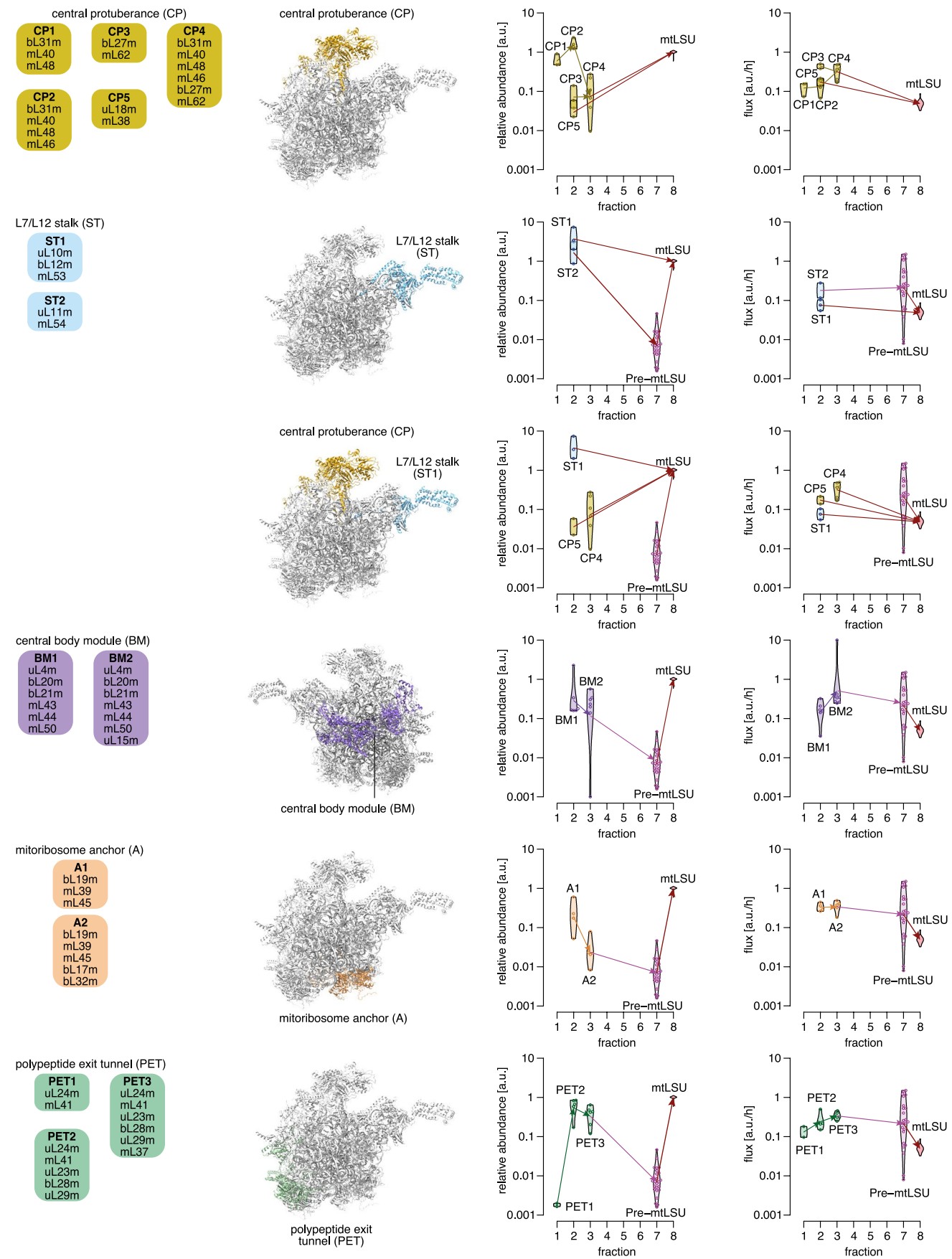

**Extended Data Fig. 6 | See next page for caption.**

**Extended Data Fig. 6 | Cluster analysis of mtLSU assembly modules.** Left-hand panel: schematic representation of the mtLSU assembly modules and their position in the mature mtLSU (PDB: 6zm6). Right-hand panel: clustering reveals 15 distinct mtLSU assembly modules. Violin plots illustrate the distribution of estimated fluxes and abundances over mtLSU MRPs that are grouped into the same assembly module depending on the sucrose gradient fraction they were detected in. Arrows indicate the further assembly direction of a submodule. Color labels correspond to left panel. ST – L7/L12 stalk; CP – central protuberance; PET – polypeptide exit tunnel; A – mitoribosome membrane anchor; BM – central body assembly module.

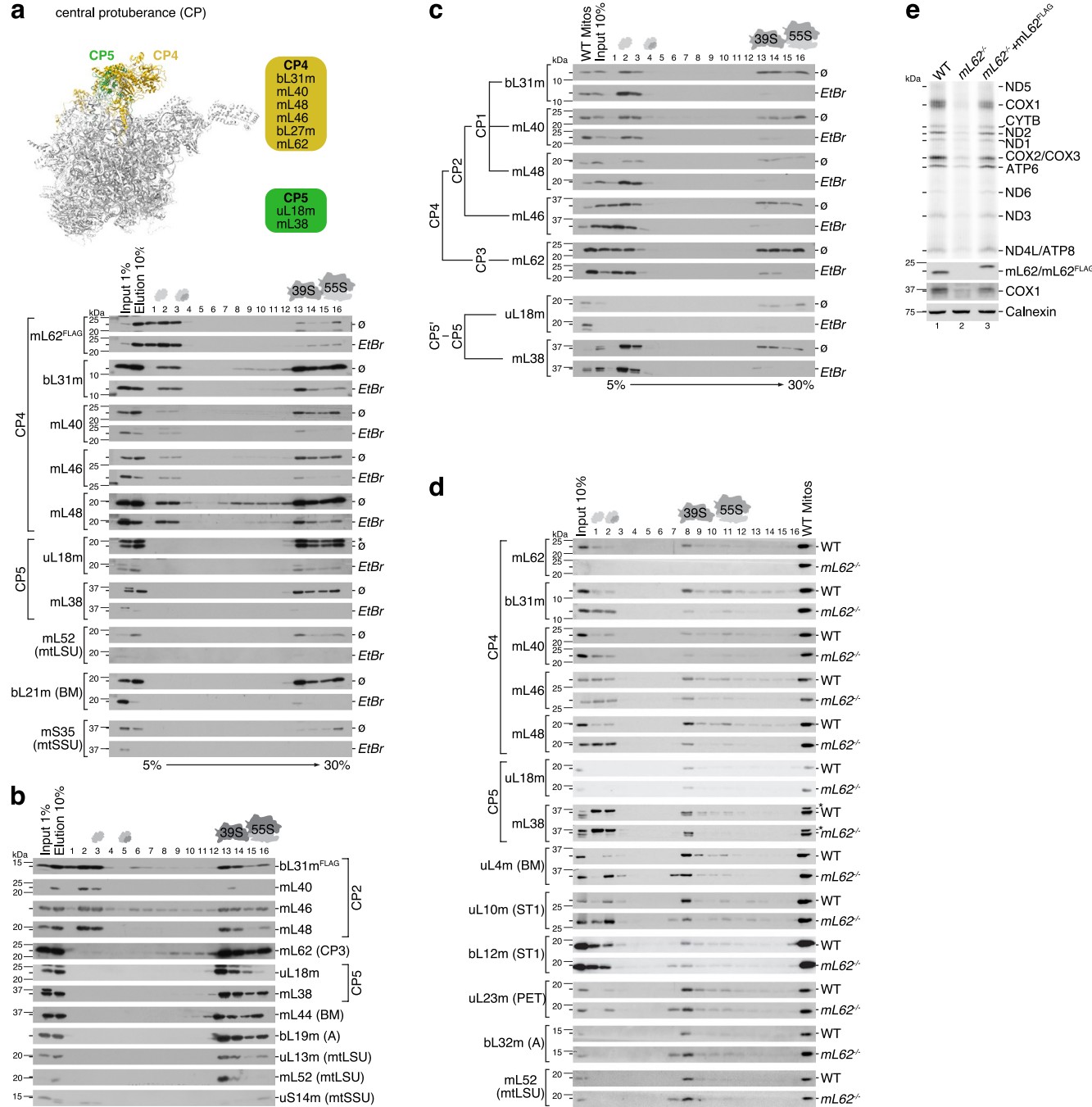

**Extended Data Fig. 7 | Assembly of the central protuberance. a**, Formation of the CP assembly module upon 16 S mt-rRNA depletion caused by ethidium bromide (EtBr) treatment; non-treated cell line (Ø) serves as a control. Mitoribosomal complexes were immuno-isolated via FLAG-tagged CP constituent mL62 and separated by sucrose gradient ultracentrifugation. Upper panel indicates the position of the CP4 and CP5 assembly modules in the mature mtLSU (PDB: 6zm6). **b**, The composition of the CP2 assembly cluster. Mitoribosomal complexes were immuno-isolated via FLAG-tagged CP constituent bL31m and separated by sucrose gradient centrifugation.

**c**, Formation of the CP cluster upon rRNA depletion. Mitoribosomal complexes isolated from ethidium bromide-treated (EtBr) or untreated cells (Ø) were separated by sucrose gradient ultracentrifugation. **d**, Sucrose gradient sedimentation of the mitoribosomal complexes formed in the absence of mL62. **e**, Mitochondrial translation in mL62-deficient cells was monitored by *de novo* incorporation of [³⁵S]Methionine and visualized by autoradiography. ST – L7/L12 stalk; CP – central protuberance; PET – polypeptide exit tunnel; A – mitoribosome membrane anchor; BM – central body assembly module.

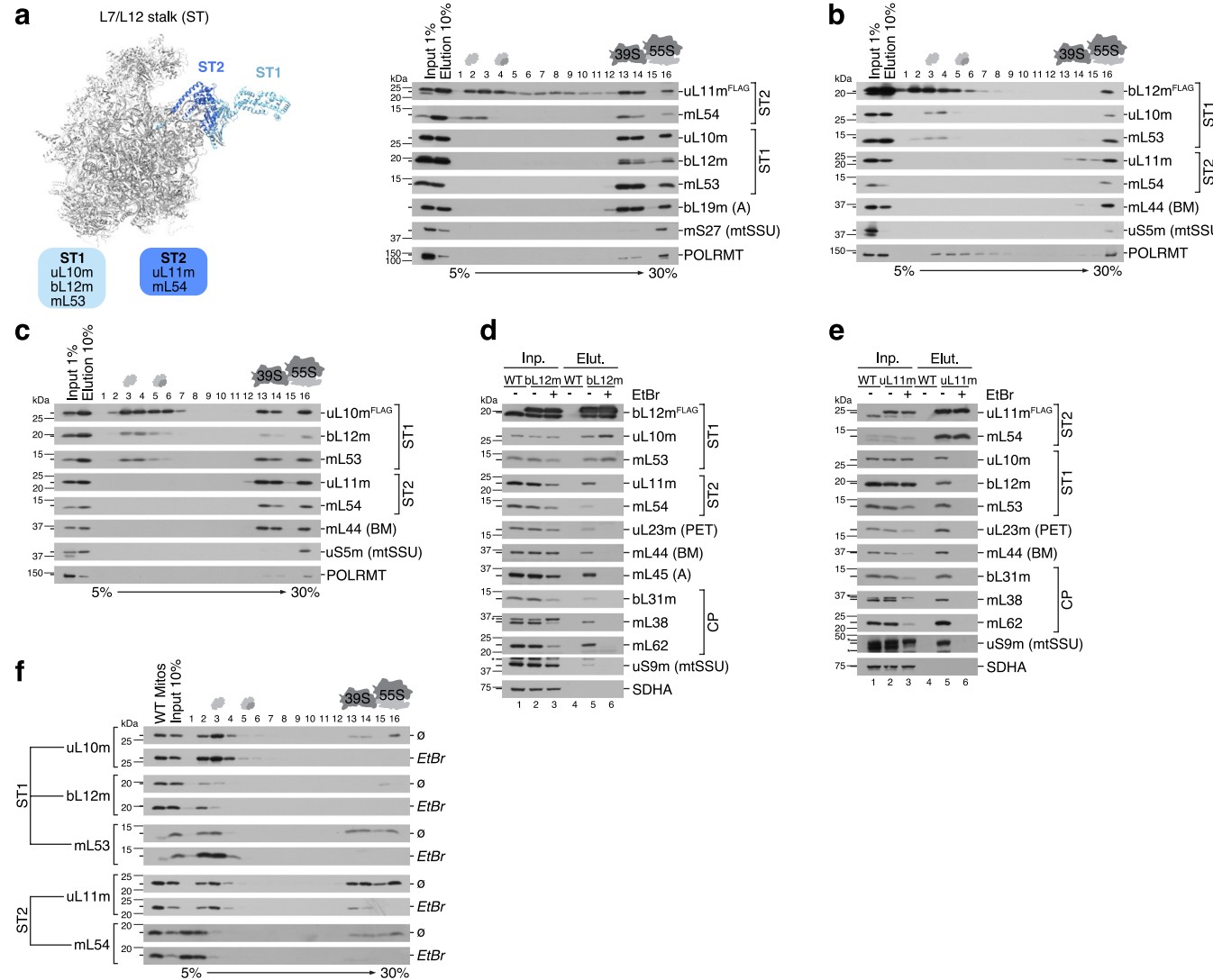

**Extended Data Fig. 8 | Assembly of the L7/L12 stalk. a-c**, The composition of the ST assembly cluster. Mitoribosomal complexes were immuno-isolated via FLAG-tagged ST constituents uL11m (**a**), bL12m (**b**) or uL10m (**c**) and separated by sucrose gradient centrifugation. Left-hand panel in (**a**) indicates the position of the ST1 and ST2 assembly modules in the mature mtLSU (PDB: 6zm6). **d-e**, Immunoisolation of ST1 and ST2 assembly modules via FLAG-tagged constituents bL12m (**d**) and uL11m (**e**), respectively in the absence of rRNA (*EtBr+*). **f**, Formation of the ST cluster upon rRNA depletion. Mitoribosomal complexes isolated from ethidium bromide-treated (EtBr) or untreated cells (Ø) were separated by sucrose gradient ultracentrifugation. ST – L7/L12 stalk; CP – central protuberance; PET – polypeptide exit tunnel; A – mitoribosome membrane anchor; BM – central body assembly module.

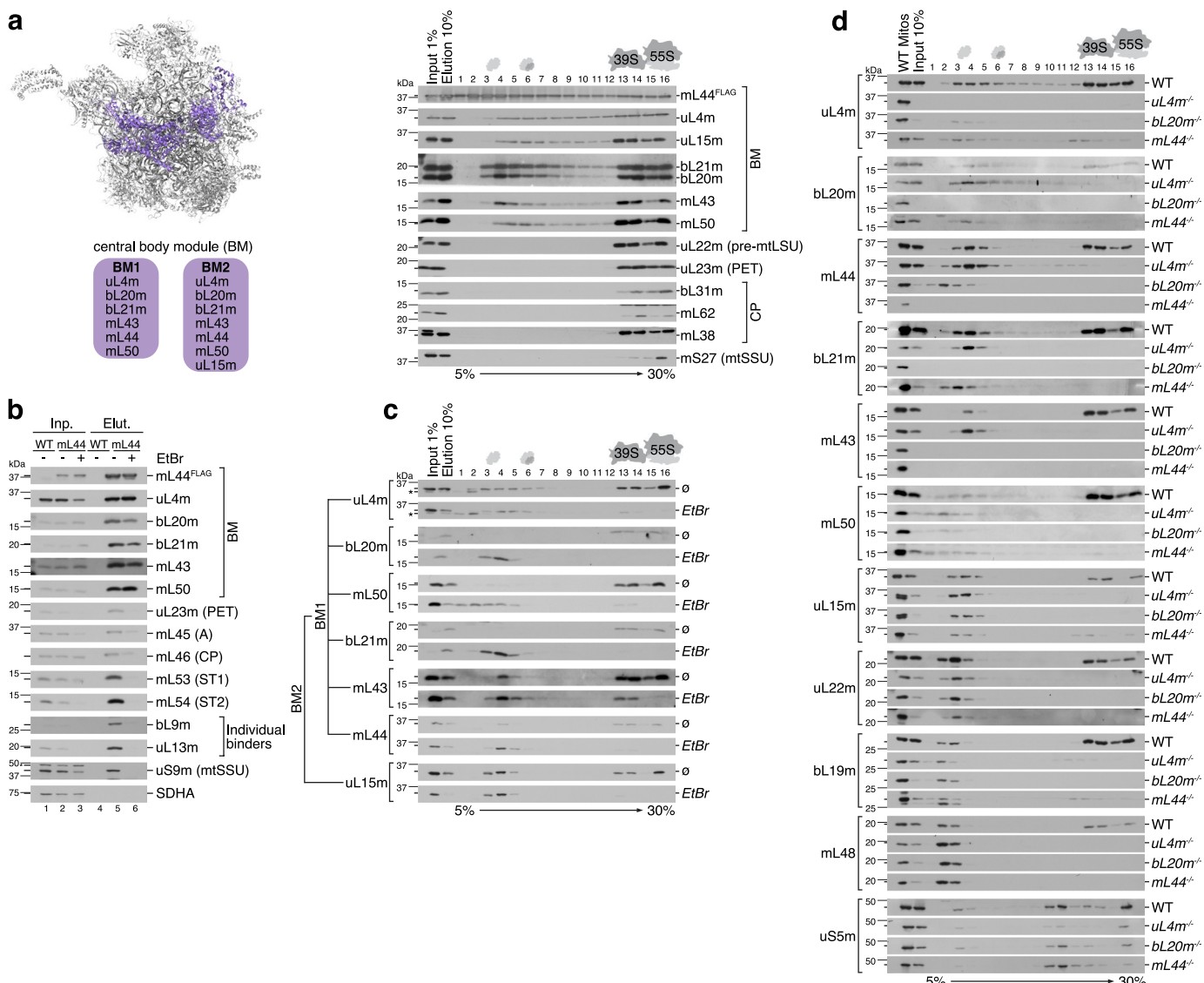

**Extended Data Fig. 9 | Formation of the mitoribosomal central body module.**
**a**, The composition of the BM assembly cluster. Mitoribosomal complexes were immuno-isolated via FLAG-tagged BM constituent mL44 and separated by sucrose gradient centrifugation. Left-hand panel indicates the position of the BM assembly module in the mature mtLSU (PDB: 6zm6). **b**, Immunoisolation of BM assembly module via FLAG-tagged constituent mL44 in the absence of rRNA (*EtBr+*). **c**, Formation of the BM cluster upon rRNA depletion. Mitoribosomal complexes isolated from ethidium bromide-treated (EtBr) or untreated cells (Ø) were separated by sucrose gradient ultracentrifugation. **d**, Sucrose gradient sedimentation analysis of mitoribosomal complexes formed in cell lines deficient for uL4m, bL20m or mL44. ST – L7/L12 stalk; CP – central protuberance; PET – polypeptide exit tunnel; A – mitoribosome membrane anchor; BM – central body assembly module.

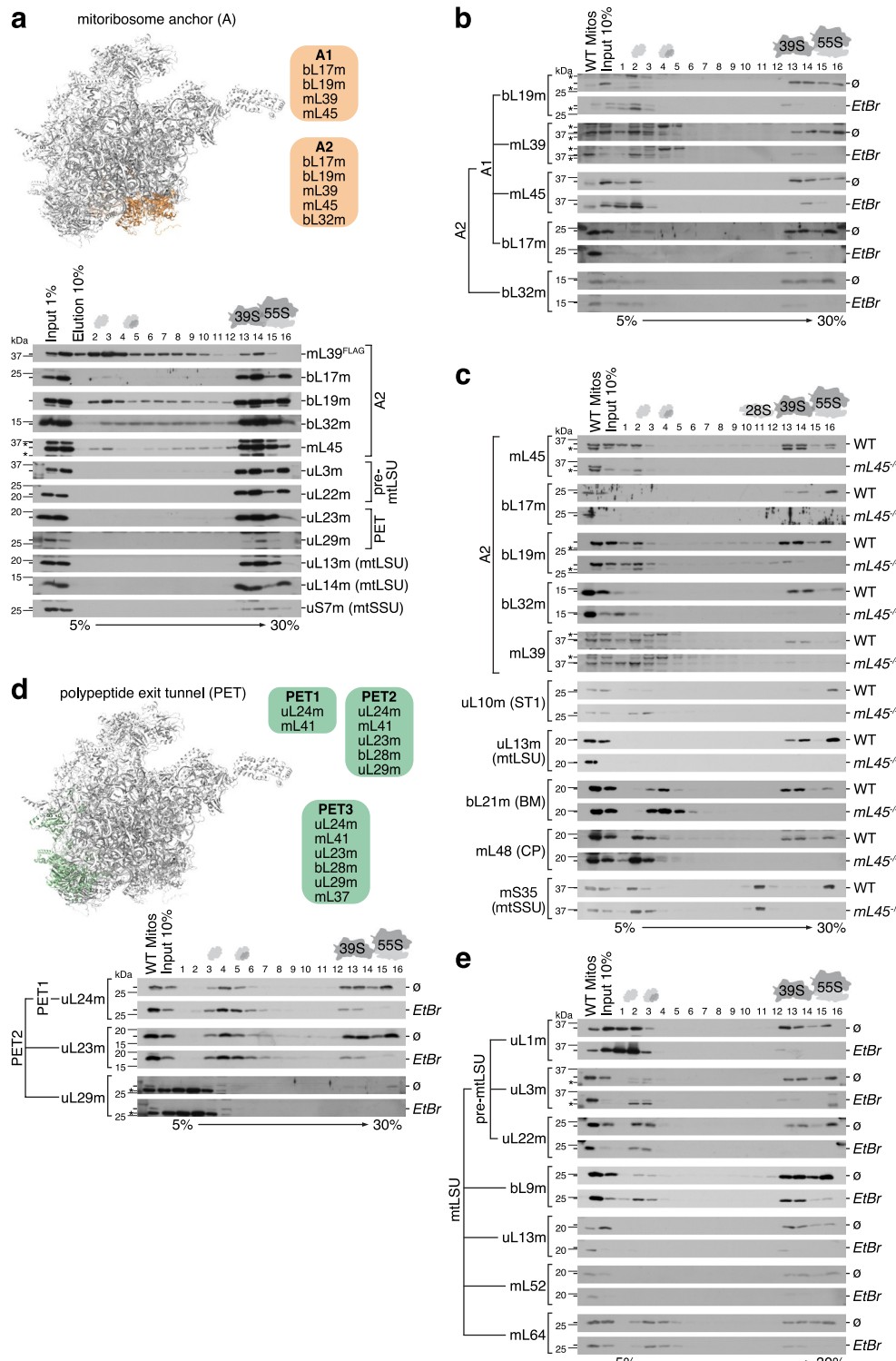

**Extended Data Fig. 10 | Biogenesis of the mitoribosome anchor and polypeptide exit tunnel modules. a**, The composition of the A assembly cluster. Mitoribosomal complexes were immuno-isolated via FLAG-tagged A constituent mL39 and separated by sucrose gradient centrifugation. mtSSU constituent uS7m serves as a negative control. Upper panel indicates the position of the A module in the mature mtLSU (PDB: 6zm6). **b**, Formation of the A cluster upon rRNA depletion. Mitoribosomal complexes isolated from ethidium bromide-treated (EtBr) or untreated cells (Ø) were separated by sucrose

gradient ultracentrifugation. **c**, Sucrose gradient sedimentation analysis of mitoribosome complexes in mL45-deficient cells. **d**, Formation of the PET cluster upon rRNA depletion was assessed as in **b**. Upper panel indicates the position of the PET assembly module in the mature mtLSU (PDB: 6zm6). **e**, Sucrose gradient sedimentation analysis of the MRPs which assemble into the mtLSU individually. ST – L7/L12 stalk; CP – central protuberance; PET – polypeptide exit tunnel; A – mitoribosome membrane anchor; BM – central body assembly module.

# Reporting Summary

## Statistics

For all statistical analyses, confirm that the following items are present in the figure legend, table legend, main text, or Methods section.

| n/a | Confirmed | |
|---|---|---|
| ☐ | ☒ | The exact sample size (*n*) for each experimental group/condition, given as a discrete number and unit of measurement |
| ☐ | ☒ | A statement on whether measurements were taken from distinct samples or whether the same sample was measured repeatedly |
| ☐ | ☒ | The statistical test(s) used AND whether they are one- or two-sided *Only common tests should be described solely by name; describe more complex techniques in the Methods section.* |
| ☒ | ☐ | A description of all covariates tested |
| ☒ | ☐ | A description of any assumptions or corrections, such as tests of normality and adjustment for multiple comparisons |
| ☐ | ☒ | A full description of the statistical parameters including central tendency (e.g. means) or other basic estimates (e.g. regression coefficient) AND variation (e.g. standard deviation) or associated estimates of uncertainty (e.g. confidence intervals) |
| ☐ | ☒ | For null hypothesis testing, the test statistic (e.g. *F*, *t*, *r*) with confidence intervals, effect sizes, degrees of freedom and *P* value noted *Give P values as exact values whenever suitable.* |
| ☐ | ☒ | For Bayesian analysis, information on the choice of priors and Markov chain Monte Carlo settings |
| ☒ | ☐ | For hierarchical and complex designs, identification of the appropriate level for tests and full reporting of outcomes |
| ☒ | ☐ | Estimates of effect sizes (e.g. Cohen's *d*, Pearson's *r*), indicating how they were calculated |

*Our web collection on statistics for biologists contains articles on many of the points above.*

## Software and code

Policy information about availability of computer code

| Data collection | for mass spectrometry data acquisition: Thermo Fisher Scientific software: Thermo Xcalibur Instrument Setup, Thermo Scientific Xcalibur Version 4. 4.16.14, Tune Application, Version 4.0.309.28 (for monitoring detection). |
|---|---|
| Data analysis | - Affinity Designer V2 (Serif Europe Ltd); Affinity Photo V2 (Serif Europe Ltd); Typhoon imaging system v. 8.1 (GE Healthcare); PDBePISA (European Bioinformatics Institute); UCSF ChimeraX (Resource for Biocomputing, Visualization, and Informatics (RBVI)); MaxQuant software version 1. 6.0 .1 (software to analyze mass spectrometry data) <br> - custom scripts: R scripts to process and visualize MaxQuant software outputs, R scripts to model mtSSU kinetics (detailed in methods; used R packages: deSolve v1.35, stringr v1.5.0, dplyr v1.1.1, vroom v1.6.1, tidr v1.1.4), Python (v3.9.17) scripts to analyse unfractionated dataset (used Python packages: PyMC v 5.7.2), Matlab (v 9.13.) to obtain an algebraic solution of the ODE systems of the one-state-model and of the two-state-model <br><br> No custom algorithm or software were utilized in this study. R Scripts and Python scripts for data processing and visualization are provided on Figshare: 10.6084/m9.figshare.25343125 |

For manuscripts utilizing custom algorithms or software that are central to the research but not yet described in published literature, software must be made available to editors and reviewers. We strongly encourage code deposition in a community repository (e.g. GitHub). See the Nature Portfolio guidelines for submitting code & software for further information.

## Data

Policy information about availability of data

All manuscripts must include a data availability statement. This statement should provide the following information, where applicable:

- Accession codes, unique identifiers, or web links for publicly available datasets
- A description of any restrictions on data availability
- For clinical datasets or third party data, please ensure that the statement adheres to our policy

A data availability statement is included in the manuscript.
Material will be available upon reasonable request and source data are provided with this paper. The
original data generated in this study are provided in the supplementary information and the source data
files.
The mass spectrometry original data, protein sequence databases (downloaded from UniProt Knowledgebase), MaxQuant analysis files, and database search output
files have been deposited via the MASSIVE repository and are available using the following identifiers: MSV000091653, MSV000091652.

## Research involving human participants, their data, or biological material

Policy information about studies with human participants or human data. See also policy information about sex, gender (identity/presentation), and sexual orientation and race, ethnicity and racism.

| | |
|---|---|
| Reporting on sex and gender | n.a. |
| Reporting on race, ethnicity, or other socially relevant groupings | n.a. |
| Population characteristics | n.a. |
| Recruitment | n.a. |
| Ethics oversight | n.a. |

Note that full information on the approval of the study protocol must also be provided in the manuscript.

# Field-specific reporting

Please select the one below that is the best fit for your research. If you are not sure, read the appropriate sections before making your selection.

☒ Life sciences  ☐ Behavioural & social sciences  ☐ Ecological, evolutionary & environmental sciences

For a reference copy of the document with all sections, see nature.com/documents/nr-reporting-summary-flat.pdf

# Life sciences study design

All studies must disclose on these points even when the disclosure is negative.

| | |
|---|---|
| Sample size | No sample-size calculation was performed. All experiments were done in vitro. Sample in this study is defined as a biological replicate, which is derived from a HEK293 cell line clone. Hence, biological replicates represent variation in experimentation rather than biological variation. The number of biological replicates (n=3) was chosen based on community standard. |
| Data exclusions | No data were excluded. |
| Replication | All experiments were done in vitro with commonly used 3 biological replicates and 2 technical replicates. All replication attempts were successful. |
| Randomization | In this study, no comparison between experimental sample groups was carried out. Hence, there was no randomization necessary. |
| Blinding | Blinding is not applicable in this study, as only in vitro experiments without any human or animal subjects were carried out. The knowlegde of the experiment subject by the person carrying out the experiment has no impact on the results in this study. |

# Reporting for specific materials, systems and methods

We require information from authors about some types of materials, experimental systems and methods used in many studies. Here, indicate whether each material, system or method listed is relevant to your study. If you are not sure if a list item applies to your research, read the appropriate section before selecting a response.

## Materials & experimental systems

| n/a | Involved in the study |
|-----|----------------------|
| ☐ | ☒ Antibodies |
| ☐ | ☒ Eukaryotic cell lines |
| ☒ | ☐ Palaeontology and archaeology |
| ☒ | ☐ Animals and other organisms |
| ☒ | ☐ Clinical data |
| ☒ | ☐ Dual use research of concern |
| ☒ | ☐ Plants |

## Methods

| n/a | Involved in the study |
|-----|----------------------|
| ☒ | ☐ ChIP-seq |
| ☒ | ☐ Flow cytometry |
| ☒ | ☐ MRI-based neuroimaging |

## Antibodies

| Antibodies used | Rabbit polyclonal anti-bS1m (Proteintech; Cat#16378-1-AP); Rabbit polyclonal anti-uS5m (Proteintech; Cat#16428-1-AP); Rabbit polyclonal anti-uS7m (Sigma-Aldrich; Cat# HPA 023007); Rabbit polyclonal anti-uS9m (Proteintech; Cat#16533-1-AP); Rabbit polyclonal anti-uS10m (Proteintech; Cat#16030-1-AP); Rabbit polyclonal anti-uS11m (Proteintech; Cat#17041-1-AP); Rabbit polyclonal anti-uS12m (Proteintech, Cat#15225-1-AP); Rabbit polyclonal anti-uS14m (Proteintech; Cat#16301-1-AP); Rabbit polyclonal anti-uS15m (Proteintech; Cat#17106-1-AP); Rabbit polyclonal anti-bS16m (Proteintech; Cat#16735-1-AP); Rabbit polyclonal anti-uS17m (ProteinTech; Cat#18881-1-AP); Rabbit polyclonal anti-mS22 (ProteinTech; Cat#10984-1-AP); Rabbit polyclonal anti-mS23 (ProteinTech; Cat#18345-1-AP); Rabbit polyclonal anti-mS25 (ProteinTech; Cat#15277-1-AP); Rabbit polyclonal anti-mS27 (ProteinTech; Cat#17280-1-AP); Rabbit polyclonal anti-mS29 (ProteinTech; Cat#10276-1-AP); Rabbit polyclonal anti-mS31 (ProteinTech; Cat#16288-1-AP); Rabbit polyclonal anti-mS34 (ProteinTech; Cat#15166-1-AP); Rabbit polyclonal anti-mS35 (ProteinTech; Cat#16457-1-AP);Rabbit polyclonal anti-mS37 (Proteintech, #11728-1-AP); Rabbit polyclonal anti-mS39 (ProteinTech; Cat#25158-1-AP); Rabbit polyclonal anti-mS40 (ProteinTech; Cat#16139-1-AP); Rabbit polyclonal anti-uL1m (homemade; provided by P. Rehling); Rabbit polyclonal anti-uL3m (Proteintech; Cat#16584-1-AP); Rabbit polyclonal anti-uL4m (Proteintech; Cat#27484-1-AP); Rabbit polyclonal anti-bL9m (Proteintech; Cat#15342-1-AP); Rabbit polyclonal anti-uL10m (Proteintech; Cat#16652-1-AP); Rabbit polyclonal anti-uL11m (Proteintech; Cat#15543-1-AP); Rabbit polyclonal anti-bL12m (Proteintech; Cat#14795-1-AP); Rabbit polyclonal anti-uL13m (Proteintech; Cat#16241-1-AP); Rabbit polyclonal anti-uL14m (Proteintech, Cat#15040-1-AP); Rabbit polyclonal anti-uL15m (Proteintech; Cat#18339-1-AP); Rabbit polyclonal anti-bL17m (Proteintech; Cat#17214-1-AP); Rabbit polyclonal anti-uL18m (Proteintech; Cat#15178-1-AP); Rabbit polyclonal anti-bL19m (Proteintech; Cat#16517-1-AP); Rabbit polyclonal anti-bL20m (Proteintech; Cat#16969-1-AP); Rabbit polyclonal anti-bL21m (Proteintech; Cat#16978-1-AP); Rabbit polyclonal anti-uL22m (Proteintech; Cat#16299-1-AP); Rabbit polyclonal anti-uL23m (homemade; provided by P. Rehling); Rabbit polyclonal anti-uL24m (Proteintech; Cat#16224-1-AP); Rabbit polyclonal anti-uL29m (Proteintech; Cat#24728-1-AP); Rabbit polyclonal anti-bL31m (Proteintech; Cat#17679-1-AP); Rabbit polyclonal anti-bL32m (homemade; provided by P. Rehling); Rabbit polyclonal anti-mL38 (Proteintech; Cat#15913-1-AP); Rabbit polyclonal anti-mL39 (homemade; provided by P. Rehling); Rabbit polyclonal anti-mL40 (Novusbio; Cat#NBP1-82620); Rabbit polyclonal anti-mL43 (Proteintech; Cat#17477-1-AP); Rabbit polyclonal anti-mL44 (Proteintech; Cat#16394-1-AP); Rabbit polyclonal anti-mL45 (Proteintech; Cat#15682-1-AP); Rabbit polyclonal anti-mL46 (Proteintech; Cat#16611-1-AP); Rabbit polyclonal anti-mL48 (Proteintech; Cat#14677-1-AP); Rabbit polyclonal anti-mL50 (Invitrogen; Cat#PA5-54638); Rabbit polyclonal anti-mL52 (Proteintech; Cat#16800-1-AP); Rabbit polyclonal anti-mL53 (Proteintech; Cat#16142-1-AP); Rabbit polyclonal anti-mL54 (Proteintech; Cat#17683-1-AP); Rabbit polyclonal anti-mL62 (Proteintech; Cat#10403-1-AP); Rabbit polyclonal anti-mL64 (Proteintech; Cat#16260-1-AP); Mouse monoclonal anti-Calnexin (Proteintech; Cat#66903-1-Ig, clone 2A2C6); Rabbit polyclonal anti-COX1 (homemade; provided by P. Rehling); Rabbit polyclonal anti-POLRMT (Proteintech; Cat#17748-1-AP). Mouse monoclonal anti-SDHA (ThermoFisher Scientific, Cat#459200, clone clone 2E3GC12FB2AE2) |
|---|---|
| Validation | Commercially supplied antibodies were validated by manufacturers by subjecting lysates of multiple human cell lines (e.g. HEK, HeLa, HepG2) or human tissue (e.g. liver) to SDS-PAGE followed by immunoblotting using the respective antibodies. Additionally, all antibodies for mitoribosomal proteins used in this study follow the expected behavior, meaning detection of ribosomal proteins i.) co-migrating with the mitoribosomal particles in sucrose gradients, and ii.) co-purifying during FLAG-immunoprecipitation of mitoribosome complexes.<br>Homemade antibodies were validated in previous studies (Rabbit polyclonal anti-uL1m, Rabbit polyclonal anti-uL23m and Rabbit polyclonal anti-COX: Richter-Dennerlein et al., 2016; Rabbit polyclonal anti-bL32m: Lavdovskaia et al., 2018). Rabbit polyclonal anti-mL39 was validated by western blot using cell line overexpressing FLAG-tagged mL39 (HEK293-Flp-In T-Rex mL39FLAG, see Extended Data Fig.10a). |

## Eukaryotic cell lines

Policy information about cell lines and Sex and Gender in Research

| Cell line source(s) | HEK293-Flp-In T-Rex (Thermo Fisher Scientific; R78007); HEK293-Flp-In T-Rex uL4m-/-, HEK293-Flp-In T-Rex bL20m-/-, HEK293-Flp-In T-Rex mL44-/-, HEK293-Flp-In T-Rex mL45-/-, HEK293-Flp-In T-Rex mL62-/-, HEK293-Flp-In T-Rex uS7m-/-, and HEK293-Flp-In T-Rex mS40-/- cell lines were generated using CRISPR/Cas9 technology; HEK293-Flp-In T-Rex uL10m-FLAG, HEK293-Flp-In T-Rex uL11m-FLAG, HEK293-Flp-In T-Rex bL12m-FLAG, HEK293-Flp-In T-Rex bL31m-FLAG, HEK293-Flp-In T-Rex mL39-FLAG, HEK293-Flp-In T-Rex mL44-FLAG, HEK293-Flp-In T-Rex mL62-FLAG, HEK293-Flp-In T-Rex bS1m-FLAG, HEK293-Flp-In T-Rex uS10m-FLAG, HEK293-Flp-In T-Rex mS22-FLAG, HEK293-Flp-In T-Rex mS25-FLAG, HEK293-Flp-In T-Rex mS27-FLAG, and HEK293-Flp-In T-Rex mS40-FLAG cell lines were generated by co-transfection of the maternal HEK293-Flp-In T-Rex WT cell line with pcDNA5/FRT/TO bearing the respective FLAG-tagged MRP nucleotide sequence and pOG44 Flp-Recombinase Expression Vector. |
|---|---|
| Authentication | HEK293-Flp-In T-Rex cell lines were routinely treated with Blasticidin S to ensure the authentic presence of the Blasticidin S-resistance locus. Stable insertion of the FLAG-tagged MRPs into the T-Rex expression cassette were confirmed by western |

blot using mouse monoclonal anti-FLAG (Sigma-Aldrich) antibody. The cell lines expressing FLAG-tagged MRPs were routinely treated with Hygromycin B to ensure the stability of the insertion. Knockout cell lines were confirmed by western blotting and Sanger gDNA sequencing.

Mycoplasma contamination — Cell Lines used in this study were systematically tested negative for the presence of Mycoplasma by GATC Biotech.

Commonly misidentified lines
(See ICLAC register) — No commonly misidentified cell lines were used.

## Plants

Seed stocks — *Report on the source of all seed stocks or other plant material used. If applicable, state the seed stock centre and catalogue number. If plant specimens were collected from the field, describe the collection location, date and sampling procedures.*

Novel plant genotypes — *Describe the methods by which all novel plant genotypes were produced. This includes those generated by transgenic approaches, gene editing, chemical/radiation-based mutagenesis and hybridization. For transgenic lines, describe the transformation method, the number of independent lines analyzed and the generation upon which experiments were performed. For gene-edited lines, describe the editor used, the endogenous sequence targeted for editing, the targeting guide RNA sequence (if applicable) and how the editor was applied.*

Authentication — *Describe any authentication procedures for each seed stock used or novel genotype generated. Describe any experiments used to assess the effect of a mutation and, where applicable, how potential secondary effects (e.g. second site T-DNA insertions, mosiacism, off-target gene editing) were examined.*

