## [Peer Review File · Nature Structural & Molecular Biology]

Peer Review Information

Manuscript Title: A roadmap for ribosome assembly in human mitochondria

Corresponding author name(s): Ricarda Richter-Dennerlein, Henning Urlaub, Juliane Liepe

Editorial Notes:

Transferred manuscripts This manuscript has been previously reviewed at another journal that is not operating a transparent peer review scheme. This document only contains reviewer comments, rebuttal and decision letters for versions considered at Nature Structural & Molecular Biology

Reviewer Comments & Decisions:

Decision Letter, initial version:
--

Message: Our ref: NSMB-A48545-T

22nd Dec 2023

Dear Dr. Richter-Dennerlein,

Thank you for submitting your revised manuscript "A roadmap for ribosome assembly in human mitochondria" (NSMB-A48545-T). It has now been assessed by the editorial team, who have found that it has improved in revision, and therefore we'll be happy in principle to publish it in Nature Structural & Molecular Biology, pending minor revisions to comply with our editorial and formatting guidelines.

We are now performing detailed checks on your paper and will send you a checklist detailing our editorial and formatting requirements in the next few weeks. Please note that delays may be expected at this time of the year. Please do not upload the final materials and make any revisions until you receive this additional information from us.

Thank you again for your interest in Nature Structural & Molecular Biology. Please do not hesitate to contact me if you have any questions.

Sincerely,

Sara

Sara Osman, Ph.D.
Associate Editor
Nature Structural & Molecular Biology

Author Rebuttal to Initial comments

Referee #2 (Remarks to the Author):

The revised manuscript by Lavdovskaia et al. has been improved in terms of text and new analyses/figures. The key finding of this manuscript was that mitochondrial ribosomal protein-only modules are initially assembled and subsequently assembled on the rRNA scaffold, which is a fundamentally different assembly pathway compared to its bacterial mitoribosomes or cytosolic ribosomes. In the first version of the manuscript, I did not fully understand what they claimed from the original datasets and thought that more careful validations must be required. In the revised manuscript, however, the key finding has been further supported by new experiments/analyses, including the EtBr treatments (Fig. 4), the one-state and two-state-models of MRPs (Extended Data Fig. 1), and clear representation of Extended Data Figs. 7-10. All new pieces support the notion that the formation of protein-only modules is independent of the presence of rRNA.

I therefore support the data and claim by Lavdovskaia et al. from a proteomics/biochemical point of view.

Our response:

We thank the reviewer for his/her positive feedback about our revised manuscript.

Referee #3 (Remarks to the Author):

The authors have addressed the reviewer comments very well in their revisions, which have greatly improved their manuscript on human mitochondrial ribosome assembly. In particular, the new Fig. 1 outlines the method and clustering process more clearly. I found the schemes throughout the figures very helpful. The high-resolution sucrose gradients, and the northern blots, now provide much more compelling support for the existence of mitochondrial ribosomal protein complexes. I am largely satisfied with that this conclusion of the paper is supported by the data in the revised manuscript.

Our response:

We are very pleased to receive this positive feedback about the restructured manuscript.

The only aspect of the interpretation that remains in doubt (to me) is whether the pulse-chase scheme can resolve the kinetics of ribosome assembly in ED Fig. 4 as sensitively as implied by the model and the discussion of the paper. I don't think this is a major conclusion of the paper, and so it is OK. The only improvement I would suggest is to downplay or remove the statement about "rate-limiting steps" in the abstract and in the discussion.

Our response:

Unfortunately, we have to disagree with this statement and we will aim to clarify this in the manuscript. Neither the kinetic model of the mtSSU assembly shown in ED Fig. 4 nor the reconstructed assembly pathways of the mtSSU and mtLSU are based on the unfractionated triple SILAC data displayed in ED Fig. 1. Latter was only used to characterise the global turnover of mtLSU and mtSSU subunits and known assembly factors inside mitochondria, as well as in whole cell lysate. This dataset provided us the insight that most likely all subunits are rapidly transported into mitochondria, as their turnover in mitochondria and in whole cell lysate did not differ.

We agree with the reviewer, that this dataset is not suited to dissect the assembly pathway and for sure it is not suited to derive precise association and dissociation rates of individual assembly steps. Indeed,

for the latter, we generated a second triple SILAC dataset, which makes use of sucrose fractionation. In this second dataset (Report Figure R3.2 and Report Figure R3.3) we (i) have a higher temporal resolution, and (ii) observe the most dynamic behaviour in the first 5 fractions, in which 'heavy' labelled MRPs decrease up to 50% in the first six hours, while 'heavy' labelled MRPs in the denser sucrose gradient fractions only decrease by max. 30%. Especially the early sucrose gradient fractions describe the interaction and turnover of single proteins and small sub-complexes. Therefore, we do think that our experimental design is suitable to construct the assembly pathway and determine rate-limiting steps. However, we will adjust the wording in the abstract as suggested by the reviewer, as this aspect is indeed not a major conclusion of the paper.

Here is why I still have some doubts on this point: The authors now use a two-state model to describe the protein turnover, which is more apt, but not well constrained by the data. It is easy to see in ED Fig. 1 that the fast phase is largely complete by 3 hours, which is the first time point. The baseline estimates of heavy proteins at time zero are also quite noisy. These factors translate to a large uncertainty for the rate constant k_{ab} , which is the likelihood of a protein transferring from state A to state B, versus turnover from state A.

Our response:

Indeed, we fully agree with the reviewer that this unfractionated dataset in ED Fig 1 is not suited to infer individual turnover and binding rates at high accuracy. Already the model itself - 2-state model - is a strong simplification to describe the mitochondrial ribosome assembly pathway, because we would expect more than two states depending on the specific protein modelled. For this reason, we did not use this dataset to infer the assembly pathway or to inform the kinetic mtSSU model (see also above).

The authors misunderstood my point about this in their reply that the chosen time points cover the protein half-lives. This is true for the stable population, but not for the unstable population that is being partitioned between assembly and turnover.

The authors might want to note that the reported binding and unbinding rate constants in their models are (likely) the composites of many smaller reaction steps. A simple association reaction should occur in minutes, not hours, even assuming a sluggish on-rate of 10^7 /Ms and a concentration of 1 nM.

Our response:

We are assuming that the reviewer refers to the parameter estimates of the kinetic mtSSU model displayed in Supplementary Fig.11. The provided estimates for on-rates are shown in A.U./h (arbitrary units per hour) and hence, can be compared among each other, but do not provide absolute estimates in absence of the precise molarity (as described in detail in supplementary methods). We must stress again, that the mtSSU kinetic model was not informed by the unfractionated data displayed in ED Fig. 1, but only by the sucrose gradient fractionated dataset.

Apart from the comment above, I think this is an interesting and well-done study that is sure to have an impact on the field of ribosome biogenesis, and on the understanding of mitochondrial physiology.

Final Decision Letter:**Message:** 17th Jun 2024

Dear Dr. Richter-Dennerlein,

We are now happy to accept your revised paper "A roadmap for ribosome assembly in human mitochondria" for publication as an Article in Nature Structural & Molecular Biology.

Your paper will be published online soon after we receive proof corrections and will appear in print in the next available issue. You can find out your date of online publication by contacting the production team shortly after sending your proof corrections.

You may wish to make your media relations office aware of your accepted publication, in case they consider it appropriate to organize some internal or external publicity. Once your paper has been scheduled you will receive an email confirming the publication

details. This is normally 3-4 working days in advance of publication. If you need additional notice of the date and time of publication, please let the production team know when you receive the proof of your article to ensure there is sufficient time to coordinate. Further information on our embargo policies can be found here: <https://www.nature.com/authors/policies/embargo.html>

Please note that *Nature Structural & Molecular Biology* is a Transformative Journal (TJ). Authors may publish their research with us through the traditional subscription access route or make their paper immediately open access through payment of an article-processing charge (APC). Authors will not be required to make a final decision about access to their article until it has been accepted. Find out more about Transformative Journals

Sincerely,
Sara

Sara Osman, Ph.D.
Senior Editor
Nature Structural & Molecular Biology